# Comparative analyses of eighteen rapid antigen tests and RT-PCR for COVID-19 quarantine and surveillance-based isolation

Chad R. Wells [1], Abhishek Pandey [1], Seyed M. Moghadas [2], Burton H. Singer[3], Gary Krieger[4,5], Richard J. L. Heron[6], David E. Turner[7], Justin P. Abshire[8], Kimberly M. Phillips [9], A. Michael Donoghue [10], Alison P. Galvani [1] & Jeffrey P. Townsend [11,12,13,14 ✉]

## Abstract

**Background** Rapid antigen (RA) tests are being increasingly employed to detect SARS-CoV-2 infections in quarantine and surveillance. Prior research has focused on RT-PCR testing, a single RA test, or generic diagnostic characteristics of RA tests in assessing testing strategies. **Methods** We have conducted a comparative analysis of the post-quarantine transmission, the effective reproduction number during serial testing, and the false-positive rates for 18 RA tests with emergency use authorization from The United States Food and Drug Administration and an RT-PCR test. To quantify the extent of transmission, we developed an analytical mathematical framework informed by COVID-19 infectiousness, test specificity, and temporal diagnostic sensitivity data. **Results** We demonstrate that the relative effectiveness of RA tests and RT-PCR testing in reducing post-quarantine transmission depends on the quarantine duration and the turn-around time of testing results. For quarantines of two days or shorter, conducting a RA test on exit from quarantine reduces onward transmission more than a single RT-PCR test (with a 24-h delay) conducted upon exit. Applied to a complementary approach of performing serial testing at a specified frequency paired with isolation of positives, we have shown that RA tests outperform RT-PCR with a 24-h delay. The results from our modeling framework are consistent with quarantine and serial testing data collected from a remote industry setting. **Conclusions** These RA test-specific results are an important component of the tool set for policy decision-making, and demonstrate that judicious selection of an appropriate RA test can supply a viable alternative to RT-PCR in efforts to control the spread of disease.

## Plain language summary

Previous research on SARS-CoV-2 infection has determined optimal timing for testing in quarantine and the utility of different frequencies of testing for infection surveillance using RT-PCR and generalized rapid antigen tests. However, these strategies can depend on the specific rapid antigen test used. By examining 18 rapid antigen tests, we demonstrate that a single rapid antigen test performs better than RT-PCR when quarantines are two days or less in duration. In the context of infection surveillance, the ability of a rapid antigen test to provide results quickly counteracts its lower sensitivity with potentially more false positives. Our findings indicate that rapid antigen tests can be a suitable alternative to RT-PCR for application in quarantine and infection surveillance.

[1] Center for Infectious Disease Modeling and Analysis (CIDMA), Yale School of Public Health, New Haven, CT, USA. [2] Agent-Based Modelling Laboratory, York University, Toronto, ON, Canada. [3] Emerging Pathogens Institute, University of Florida, Gainesville, FL, USA. [4] NewFields E&E, Boulder, CO, USA. [5] Skaggs School of Pharmacy and Pharmaceutical Science, , University of Colorado Anschutz Medical Campus, Aurora, CO, USA. [6] BP Plc, 1 ST James's Square, London, UK. [7] BP America Inc, Houston, TX, USA. [8] HSE Specialties, BHP Petroleum, Houston, TX, USA. [9] BHP Petroleum, Houston, TX, USA. [10] Group HSE, BHP Group Ltd, Melbourne, VIC, Australia. [11] Department of Biostatistics, Yale School of Public Health, New Haven, CT, USA. [12] Department of Ecology and Evolutionary Biology, Yale University, New Haven, CT, USA. [13] Program in Computational Biology and Bioinformatics, Yale University, New Haven, CT, USA. [14] Program in Microbiology, Yale University, New Haven, CT, USA. ✉email: Jeffrey.Townsend@yale.edu

Testing for SARS-CoV-2 infections has played a central role in combating the COVID-19 pandemic. Despite vaccination, testing will continue to be essential for screening and surveillance[1–3], enabling timely detection of new variants and isolation of infected individuals to reduce the risk of further disease spread. Additionally, testing can inform quarantine strategies and sufficient durations to alleviate onward transmission. For instance, previous studies have shown that a 14-day quarantine with no testing for a close contact of a case can safely be shortened to seven days if an RT-PCR test is conducted on exit from the quarantine[4–6]. Implementation of this shortened quarantine for close contacts of an identified case is now specified by the Centers for Disease Control and Prevention (CDC)[7]. Complementary analyses have also evaluated the optimal frequency of RT-PCR serial testing in at-risk populations to minimize the probability of an outbreak[8–18].

Throughout the pandemic, the diversity of SARS-CoV-2 tests with regulatory approval has increased immensely. However, there has been some dispute surrounding the utility of rapid antigen (RA) tests in infection screening and control efforts[19–24]. Although the RT-PCR tests remain the gold standard for diagnosis, RA tests have aided in scaling up testing capacities worldwide. The fast turnaround time, wider availability, and lower costs make RA tests an attractive choice for workplace screening, especially in remote environments (e.g. offshore shift teams). The RA test kits require minimal training and can be self-administered, without requiring substantial ongoing equipment maintenance and calibration.

Many businesses and organizations are shifting to using RA tests for screening employees instead of solely relying on the more costly and time-consuming RT-PCR[25–28]. Evaluation of the performance of serial RA testing in identifying cases has occurred in both the health care[29,30] and university setting[31,32]. These studies conducted screening during an active COVID-19 outbreak[31] or in a tertiary hospital setting[30]. Outside these settings, screening asymptomatic individuals without known or suspected exposure to SARS-CoV-2 has been proposed and discussed in the literature[33,34]. In remote industrial settings, exposure occurs predominantly within the isolated population, and there are distinct challenges that differ from the healthcare and university environment. Specifically, there are logistical constraints to imposing isolation or offering treatment. Therefore, we present serial testing data for an offshore oil site as there is currently minimal published evidence of the effectiveness of large-scale serial RA testing in mitigating outbreaks within an industrial setting.

One concern with RA tests is their higher rate of false positives and negatives compared to RT-PCR[21,22,35]. As a tool for workplace screening or community surveillance, testing frequency is critical to avoiding an outbreak (i.e., attaining an effective reproduction number $R_E$ that is below one)[9,35]. However, increasing the number of tests used in screening increases costs and elevates the number of false positives obtained. False-positive results do not entail direct epidemiological risks, but do lead to undesirable logistical and cost challenges. For example, in an offshore and or remote workplace setting, a false positive could necessitate medical evacuation via helicopter or other aviation platforms. From a workplace risk-analysis perspective, a false positive is less disruptive to operations than false negatives resulting in transmission and a full-scale outbreak.

There has been extensive analysis and evaluation of the optimal strategies for both RT-PCR and RA testing to mitigate SARS CoV-2 transmission[4–6,8–18,36,37]. However, most of these analyses do not quantify the degree to which their use suppresses individual-level transmission in applications of quarantine[36] or serial surveillance testing with isolation of positives[10,11,13,15,16],

and most address a generalized or single RA test[5,6,9,11,12,16,18,36,37], or focus only on RT-PCR[4,8,10,13,15,17]. In contrast, multiple RA tests have received regulatory Emergency Use Authorization (EUA)[38], each with distinct temporal diagnostic sensitivity. Although past research has compared the properties of different RA tests, these temporal differences in diagnostic sensitivity have yet to be evaluated on a daily basis since infection[39,40]. These distinct diagnostic properties could produce pronounced changes in reducing onward transmission for testing strategies previously determined when using RT-PCR testing[33].

Here we construct the temporal diagnostic sensitivity curves for 18 RA tests using data on percent positive agreement (PPA) with an RT-PCR test and temporal diagnostic sensitivity of an RT-PCR test. To determine when these RA tests can serve as a suitable alternative to the more costly and laborious RT-PCR tests, we calculated (i) their associated probabilities of post-quarantine transmission (PQT) for quarantine durations from one to 14 days with testing on exit or both entry and exit upon random entry into quarantine; (ii) their extents of onward transmission for serial testing conducted every day to every 14 days; and (iii) their associated probabilities of false-positives during serial testing. We further evaluated the utility of RA tests using data collected from two offshore oil companies in the context of quarantine and serial testing within an industrial environment.

## Methods

**Infectivity profile**. The infectivity profile was generated by fitting a Gamma distribution to observed generation times for the Delta variant[41]. Specifying a fixed duration of the incubation period of 4.4 days[41], infected individuals were considered infectious no longer than 20 days after symptom onset[42–44]. We calculated results based on infected individuals producing an average of 3.2 secondary infections in the absence of self-isolation[41].

**Diagnostic sensitivity of the RT-PCR test**. To construct a temporal diagnostic sensitivity curve, we fitted a log-Normal distribution to nasopharyngeal RT-PCR testing percent-positivity data from Hellewell et al.[11] using a maximum-likelihood approach (Supplementary Methods; Supplementary Table 1; Supplementary Fig. 1).

**Diagnostic sensitivity of antigen tests**. We fitted a linear logit model to the discrete PPA data to estimate a continuous PPA curve from the time of symptom onset for each RA test. (Supplementary Data 1–2; Supplementary Methods). PPA data were only available for post-symptom onset. Therefore, we inferred the pre-symptomatic diagnostic sensitivity of the RA tests by constructing a mapping between the inferred diagnostic sensitivity post symptom onset and the level of infectivity, then applying that mapping to pre-symptomatic levels of infectivity (Supplementary Methods)[4]. The sensitivity of a RA test was calculated as the product of the PPA curve and the diagnostic sensitivity of a RT-PCR test at the specified times (Supplementary Figs. 2–22).

For our baseline results, we examined the five most commonly used RA tests: LumiraDx, Sofia, BinaxNOW, BD Veritor, and CareStart[45]. For the analysis of the LumiraDx and CareStart antigen test, we utilized the PPA data for the anterior nasal swab, as this method of sampling was used in gathering data for the BD Veritor, BinaxNOW and Sofia antigen tests. Furthermore, the anterior nasal sample can be obtained by a broad range of individuals with less specialized training compared to a nasopharyngeal sample[46–48]. We also examined both the anterior

nasal and nasopharyngeal swab for the LumiraDx and CareStart antigen test (Supplementary Data 1–2).

We compared the PPA datasets submitted to the U.S.A. FDA with those obtained from independent studies that were conducted in a real-world setting. Specifically, we considered the independent studies for BinaxNOW[49], CareStart[50], and Sofia[32]. Both the BinaxNOW and CareStart studies were conducted at a community testing site, where the trained site collector obtained the samples[49,50]. The study for Sofia was conducted in a university setting, where the samples informing the PPA of the RA test with RT-PCR were from the university in which medical professionals conducted the swabbing[32].

**Probability of post-quarantine transmission**. Specifying 35.1% of infections are asymptomatic[51] and isolation upon symptom onset, we quantified the effectiveness of quarantine and testing strategies in reducing PQT by calculating the probability of PQT for individuals entering quarantine randomly—and not identified through contact tracing—in the absence of symptoms[4]. Accounting for substantial variance in transmission among COVID-19 cases[4,52–56], we specified that secondary cases were negative-binomially distributed:

$$f(x|k,p) = \frac{\Gamma(k+x)}{\Gamma(k)\Gamma(x+1)} p^k (1-p)^x,$$

with dispersion parameter $k = 0.25$[4,56] and $p = k / (k + R)$—such that the average number of secondary cases is equal to the expected PQT, denoted $R$. This value for the dispersion parameter is consistent with estimates from other studies[52–55]. Accordingly, the probability of PQT was calculated as $1 - f(0|k, p)$.

For quarantine durations varying from one to 14 days, we compared the probability of PQT when performing a single RT-PCR test on exit to a RA test on exit or RA tests on both entry and exit. The objective of the additional RA test on entry to the one on exit is to compensate for the reduced diagnostic sensitivity of a single RA test compared to RT-PCR.

**Probability of a false-positive**. With a specificity $\zeta_i$ for test $i$ (Supplementary Data 3) and testing every $f$ days, we computed the average probability that at least one false-positive occurred over a two-week period $P_f = \frac{1}{f}\sum_{j=1}^{f} 1 - \zeta_i^{\tau_j}$, where $\tau_j$ is the number of tests to occur in the $j^{\text{th}}$ two-week period since the start of serial testing. For each testing frequency $f$ (i.e., the time between two consecutive tests), we investigated the sequence of test times $\{1,1+f,1+2f,...,1+13f\}$ that comprises all the unique testing patterns possible over a two-week period to calculate the average probability. Defining RT-PCR to be the gold standard for testing accuracy, the specificity of a RA test was estimated as the specificity of the RT-PCR test multiplied by the percent negative agreement of the RA test with RT-PCR. This calculation provides a lower bound for the RA test specificity, given the possibility of a false-negative RT-PCR test.

**Scenario analyses**. We conducted scenario analyses to determine the impact of (i) the incubation period; (ii) the reduced diagnostic sensitivity of RA tests for low-levels of infectivity; (iii) the proportion of asymptomatic infections and the basic reproduction number; and (iv) the RT-PCR diagnostic sensitivity curve on the onward transmission after quarantine and during serial testing.

To evaluate the robustness of our results to a potentially longer duration of incubation than 4.4 days[41], we evaluated the impact of an alternate incubation period of 5.72 days and the infectivity profile associated with that longer duration[37]. A positive RA test indicates active infection, suggesting that there is a substantial concentration of virus within the sample site. Studies have

indicated that SARS-CoV-2 cannot be successfully cultured 10 days after symptom onset[44]. Thus, it is possible at some stages of the disease that a RA test will return a negative while RT-PCR still yields a positive. There is also uncertainty in the inferred diagnostic sensitivity in the later stages of disease, as the PPA was extrapolated past the last recorded data point for all RA tests. To address whether differences in the results of these tests among low levels of infectivity make a difference to our findings, we imposed a threshold on the level of infectivity below which the RA test will only return a negative: if the infectivity at the time of the RA tests was below the infectivity at 10 days post-symptom onset, then the RA test returned a negative result (Supplementary Fig. 23). We also examined a stricter threshold based on the infectivity at 5.6 days post-symptom onset (where there is 1% infectivity remaining). To assess the sensitivity of our results to variation in the proportion of asymptomatic infection and the basic reproduction number, we conducted a grid analysis with values ranging from 0.1–0.95 to 1.05–5, respectively. Furthermore, we evaluated an alternative model for the temporal RT-PCR diagnostic sensitivity curve, substituting a log-Student's-$t$ distribution for the log-Normal distribution used in the baseline analysis.

**Paired testing of oil workers in quarantine**. The BD Veritor kit was selected for use by BP/BHP because at the time it was possible to secure a sufficient volume of the test kits and readers for a potential six-month evaluation. With informed consent from the offshore oil workers, onshore paired testing was conducted by laboratory-based RT-PCR and RA testing on entry into quarantine (day 0), day three, and day four. For both RA and RT-PCR testing, swabbing within quarantine was conducted by medical personnel. The study period spanned 22 November 2020 to 17 January 2021, a period that included an observed rise in community transmission across Texas and Louisiana, the states of residence for the majority of the offshore platform workers. Before entry into quarantine, workers had to pass a pre-screening questionnaire that filtered symptomatic individuals and those with recent exposure. During quarantine, a positive RT-PCR led to removal of the individual from the quarantine environment, placing them in isolation for 10 days with medical follow-up. Workers were then approved to return to work after two negative RT-PCR tests.

**Offshore serial rapid antigen testing**. Implementation of a test-and-fly protocol began at the start of the staged vaccination rollout in the USA—2 March 2021 to 24 May 2021. Thus, none of the offshore workers were vaccinated nor likely to be vaccinated for several months. The test-and-fly-strategy consisted of initial screening for symptoms, an RT-PCR test on entry to a quarantine of approximately 24 h. The exact quarantine duration was dependent on the commercial laboratory-based RT-PCR turn-around time: after receiving a negative RT-PCR test, the worker was permitted to enter the offshore work environment. With an RT-PCR positive test, the individual was removed from the quarantine environment, and advised to conduct a 10-day home-based self-isolation with medical follow-up.

Upon entering the offshore work environment, the worker underwent serial antigen testing within their first nine of 14 or more days being offshore. Testing of a worker occurred with their informed consent either on days three, six, and nine; or on days two, five, and eight. The two patterns were selected based on the results from our data-driven model for the degree of suppression achieved at each frequency of testing, by the BD Veritor test kit (Supplementary Fig. 4). All testing was conducted by the platform medic using the BD Veritor kit and reader. Any positive

individual was isolated pending helicopter transfer (typically within 12–24 h) to the established onshore medical facility, whereupon an RT-PCR nasal swab was obtained and sent to a commercial laboratory. A positive antigen test was considered to be a false positive if the follow-up RT-PCR was negative.

Conducting this study of the onshore and offshore testing of the oil platform employees, and the use of the resulting data, was approved by the Human Participants Review Sub-Committee, York University's Ethics Review Board (Certificate Number: 2021–003). All employees that participated in the study provided informed consent. Ethical approval was not required for the datasets of the RT-PCR diagnostic sensitivity, test specificity, and the PPA for each RA test because they are available in the public domain.

**Estimating the positive agreement between BD Veritor and RT-PCR during quarantine**. We calculated the probability that an infected individual would test positive with both the RA test and RT-PCR on entry, day three of quarantine, and day four. This calculation accounts for isolation both upon symptom onset and upon a positive RT-PCR test.

**Effectiveness of the offshore serial rapid antigen testing**. To determine the number of infectious individuals that were in the offshore environment, we used our model framework for serial testing, the estimates for the specificity and sensitivity of BD Veritor from the FDA reports, the number of false positives and false negatives from the RA test, RT-PCR confirmed positives, and the probability of onward transmission from these cases using a maximum-likelihood approach.

**Reporting summary**. Further information on research design is available in the Nature Research Reporting Summary linked to this article.

## Results

RA tests were classified into three categories in terms of their PPA over time: (i) stable (Clip COVID, Liaison anterior nasal and nasopharyngeal swab, Omnia, Vitros, and Sofia), (ii) gradually declining (CareStart anterior nasal swab, LumiraDx anterior nasal swab, SCoV-2 Ag Detect, Status COVID-19/Flu, Simoa, and BinaxNOW), and (iii) rapidly declining (CareStart nasopharyngeal swab, LumiraDx nasopharyngeal swab, BD Veritor, Celltrion DiaTrust, Sofia 2 Flu+SARS, and Ellume; Supplementary Figs. 2–19; Supplementary Data 2). Among the five most commonly used RA tests (i.e., LumiraDx, Sofia, BinaxNOW, BD Veritor, and CareStart)[45], there were negligible differences in the estimated diagnostic sensitivity of LumiraDx and Sofia tests over time compared to the sensitivity of an RT-PCR test (Fig. 1). BinaxNOW, BD Veritor, and CareStart exhibited noticeably lower diagnostic sensitivity than the RT-PCR tests. We found that the diagnostic sensitivity of BD Veritor was higher than Binax-Now and CareStart around the time of symptom onset, but BinaxNow and CareStart exhibited a greater probability of detection earlier and later in the disease time course.

**Quarantine and testing strategies**. As a baseline reference for PQT, the CDC had recommended a seven-day quarantine with a RT-PCR test conducted within 48 h of quarantine exit[57]. In our analysis, the RT-PCR test was specified to be conducted 24 h before exit from quarantine, whereas no lag was specified for the RA tests. The probability of PQT after a seven-day quarantine with a negative RT-PCR test on exit was estimated to be 0.0022 (95% Credible Interval [CrI]: 0.0020–0.0027). We found that a probability of PQT equivalent to the CDC-suggested quarantine

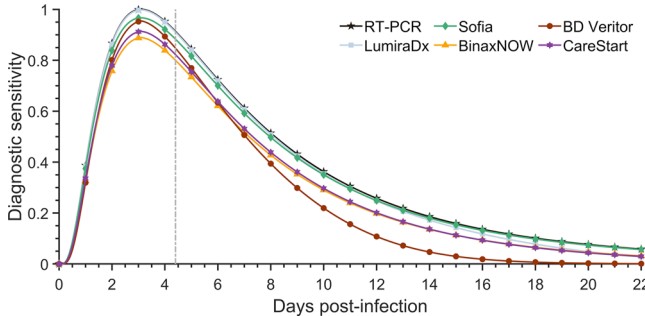

**Fig. 1 Diagnostic sensitivity of RT-PCR and rapid antigen tests.** Specifying an incubation period of 4.4 days (to symptom onset; vertical dashed line), the estimated diagnostic sensitivity of an RT-PCR test based on data from Hellewell et al.[11] (black stars) and rapid antigen test LumiraDx (light-blue squares); Sofia (green diamonds); BinaxNOW (yellow triangles); BD Veritor (dark-red circles); and CareStart (purple hexagram) based on percent positive agreement data from Emergency Use Authorization documentation over the course of infection.

and testing strategy could be achieved using RA tests with at most a one-day increase to the quarantine duration (Fig. 2a, b; Supplementary Data 4). Equivalent durations to a seven-day quarantine with an RT-PCR test conducted 24 h before exit ranged from six to seven (95% CrI: 6–7) days when a RA test was conducted on both entry and on exit from quarantine (Supplementary Data 4, Supplementary Figs. 2b–19b).

The zero-day delay in turnaround time for RA test results has the potential to offset their lower sensitivity compared to RT-PCR with a 24-h delay in obtaining results—especially in the context of short quarantine duration. Conducting a RA test on exit for quarantines with a duration of two days or shorter was more effective than an RT-PCR on exit with a 24-h delay (Fig. 2c, Supplementary Figs. 2b–19b). For over half of the RA tests, testing on both entry and exit produced a lower probability of PQT than RT-PCR on exit for quarantines of 14 days or shorter (Fig. 2d, Supplementary Figs. 2b–19b).

**Serial testing**. Incorporating a 24-h delay for obtaining RT-PCR test results, we estimated that an RT-PCR test every day was required to maintain $R_E < 1$ (Fig. 3a, b). The frequency of RA testing required to maintain $R_E < 1$ ranged from every two to every three days (Fig. 3a, Supplementary Data 4). Evaluating the importance of turnaround time for RT-PCR tests, we found that any substantial delay has a sizable impact on the required frequency of serial testing. Specifying no delay in turnaround time, RT-PCR tests can be performed just once every three (95% CrI: 2–3) days. Turnaround times for RT-PCR are highly variable both in the USA and internationally with a 12–24 hour delay being common in the USA; however, delays of up to five days have been experienced by some companies in some non-USA locations. A 24-h turnaround time leads to an $R_E$ of 0.88 (95% CrI: 0.83–0.99) when testing every day, while a 48-h delay results in an $R_E$ of 1.54 (95% CrI: 1.48–1.63; Fig. 3b).

For all testing frequencies examined, we found that RA tests on average would yield a lower $R_E$ than RT-PCR tests with a 24-h delay before receiving results (Fig. 3c). Testing every two days and isolating upon symptom onset, the average reduction in $R_E$ among the 18 RA tests was 77.9% (95% CrI: 72.8–79.2%), whereas for RT-PCR with a 24-h delay, the reduction in $R_E$ was 61.3% (95% CrI: 57.3–62.9%; Fig. 3c). All RA tests were found to maintain $R_E < 1$ when conducted at a frequency of at least once every two days (95% CrI: 1–2). However, no RA test could achieve $R_E < 1$ when the testing was less frequent than once every

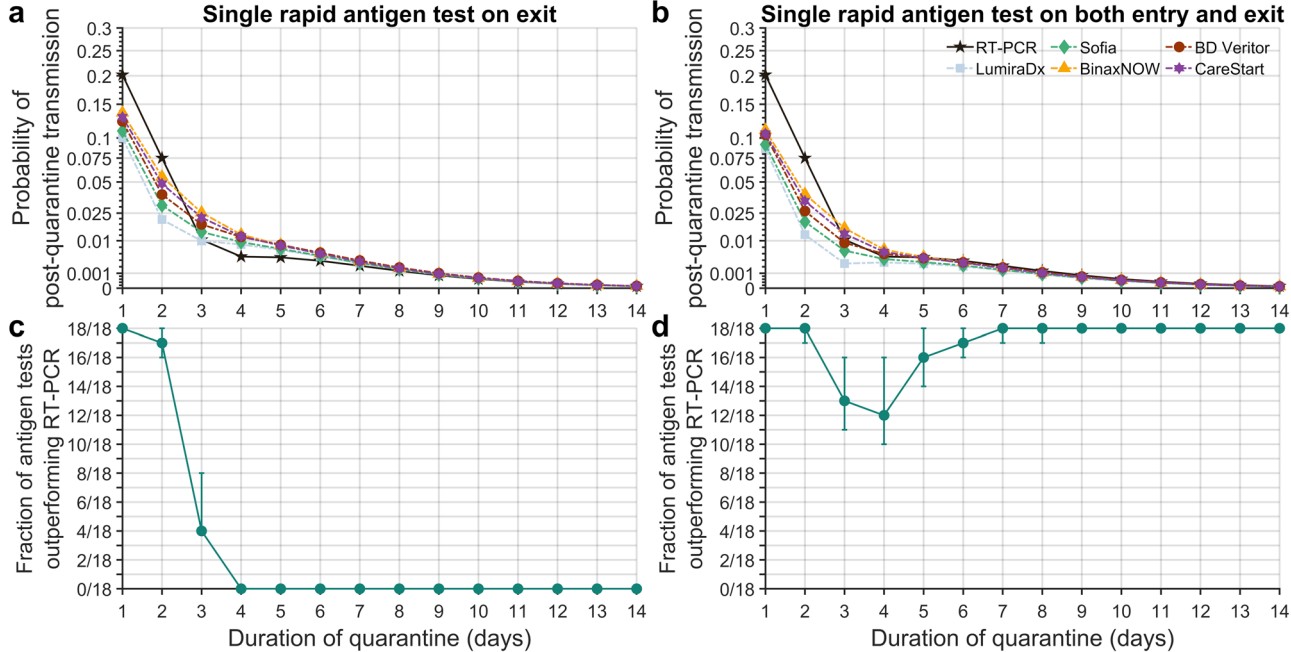

**Fig. 2 Probability of post-quarantine transmission.** Specifying a negative-binomial distribution for expected post-quarantine transmission, 35.1% of infections being asymptomatic, a 24-h delay in obtaining RT-PCR test results, no delay in receiving rapid antigen test results, an incubation period of 4.4 days, self-isolation upon symptom onset, and the diagnostic sensitivity curve for RT-PCR based on data from Hellewell et al[11], the probability of post-quarantine transmission when conducting an RT-PCR test only on exit (solid line; black stars) and the rapid antigen tests (dashed lines) LumiraDx (light-blue squares); Sofia (green diamonds); BinaxNOW (yellow triangles); BD Veritor (dark-red circles); and CareStart (purple hexagram) performed **a** on exit and **b** on both entry and exit; and the fraction of the 18 rapid antigen tests whose use conferred a lower probability of post-quarantine transmission than did an RT-PCR test conducted 24 h before exit from quarantine, when the rapid antigen test was conducted **c** on exit and **d** on both entry and exit. The whiskers denote the 95% credible interval based on 1,000 samples of an RT-PCR diagnostic sensitivity curve and rapid antigen percent positive agreement curve conducted through importance sampling.

three days (95% CrI: 3–4; Fig. 3d, Supplementary Data 4, Supplementary Fig. 24). From a logistical standpoint, RA tests are a more practical tool than RT-PCR tests for serial testing, notwithstanding false-positive rates.

**False positives for serial testing.** As the frequency of serial testing increases, an additional consideration to the reduction of transmission is the concomitant probability of obtaining false positives. For the purposes of our evaluation, we define the serial-testing false-positive rate as the probability that one or more tests yield false positives over a two-week period at the minimum frequency of testing required to maintain $R_E < 1$. Our calculations revealed an inverse relationship between $R_E$ and the probability of obtaining at least one false-positive result (Fig. 4 and Supplementary Fig. 25; Pearson correlation $r = -0.475$ and $P = 2.34 \times 10^{-16}$). For a specified $R_E$, RT-PCR tests (with at most a 48-h delay) yielded a lower probability of false-positive results than eight of the 18 RA tests (Fig. 4 and Supplementary Fig. 25). These eight higher false-positive RA tests include BD Veritor, BinaxNOW, and LumiraDx, which are among the five that are most frequently used. The serial testing false-positive rate ranged from 0.0045–0.155 among the 18 RA tests (Supplementary Data 4), whereas it was 0.013 (95% CrI: 0.004–0.022) for RT-PCR testing with a 24-h delay. Among the 18 RA tests, false-positive rates clustered into two groups: a group of eight whose serial testing false-positive rates (ranging from 0.0355 to 0.155) markedly exceeded RT-PCR (Supplementary Fig. 25, Supplementary Data 4), and 10 that exhibited markedly lower serial testing false-positive rates (but for which the upper bound of the 95% credible interval still exceeded the serial testing false-positive rate for RT-PCR with a 24-h delay; Supplementary Data 4).

**Scenario analyses.** Compared to a base-case incubation period of 4.4 days, an alternative incubation period of 5.72 days with an infectivity profile that peaks 2.4 days later (Supplementary Fig. 26) yielded somewhat contrasting results (Supplementary Data 5 vs 4). In this alternative scenario, a sole RT-PCR test on exit from quarantines of at least six days outperformed all RA tests conducted on entry to and exit from quarantine (Supplementary Fig. 27). This improved performance of RT-PCR for the alternative incubation period can be attributed to a lower average probability of identifying a case in the incubation period (0.487, 95% CrI: 0.421–0.521; vs 0.704, 95% CrI: 0.609–0.724) and to the greater proportion of transmission occurring after the incubation period (44.9% vs 37.1%). With the reduced ability in general to detect a case early in the disease time course, the addition of a RA test upon entry into quarantine provides only a marginal reduction to the PQT. For six of the 18 RA tests, the alternative incubation period necessitates an increase in frequency of testing (from every three days to every two days) to maintain $R_E < 1$ (Supplementary Datas 5 vs 4; Supplementary Figs. 28 vs 24; Supplementary Figs. 29 vs 25).

Our sensitivity curves for the RA tests featured very small but nonzero probabilities of false positive results very late in the disease time course. To assess whether these small but nonzero probabilities are having a major effect on our results, we examined a scenario in which the RA test only returned negative results at levels of infectivity below the infectivity estimated at 10 days after symptom onset. This change had no effect on our results (Supplementary Data 6). To explore the impact of an even narrower period in which a RA test can return a positive result, we imposed a higher threshold (i.e., the level of infectivity on 5.6 days after symptom onset) below which a negative result was

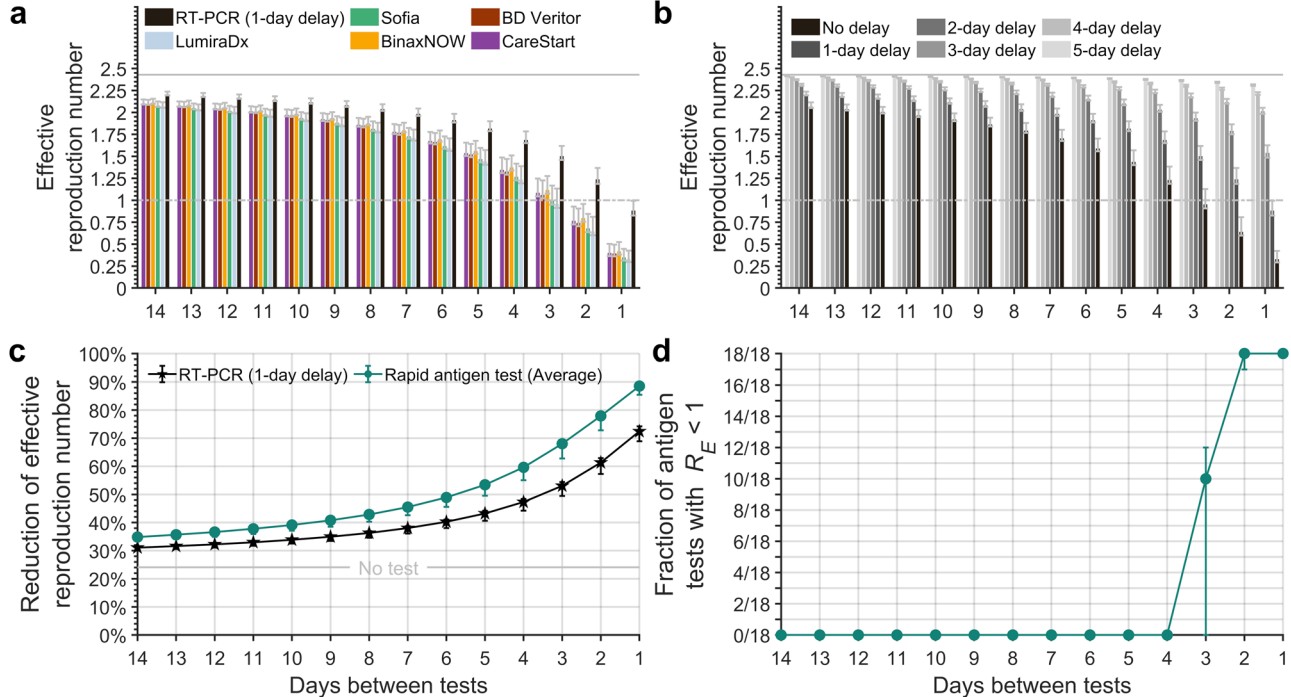

**Fig. 3 Effective reproduction number in the context of different frequencies of serial testing.** Specifying 35.1% of infections as asymptomatic, a 24-h delay in obtaining RT-PCR test results, no delay in receiving rapid antigen test results, an incubation period of 4.4 days, self-isolation upon symptom onset, and a RT-PCR diagnostic sensitivity curve informed by data from Hellewell et al.[11], **a** the expected effective reproduction number with serial testing using an RT-PCR test (black) and the rapid antigen tests LumiraDx (light-blue); Sofia (green); BinaxNOW (yellow); BD Veritor (dark-red); and CareStart (purple); and **b** for serial testing every day to every 14 days with a zero- to five-day delay (black to light gray) in obtaining the results for an RT-PCR test and isolation of positives in comparison to no testing (solid gray line). **c** The mean reduction of the effective reproduction number (with no self-isolation upon symptom onset) of the 18 RA tests (green circles), of RT-PCR with a one-day delay (black stars), and no test (gray line), with self-isolation upon symptom onset. **d** The fraction of rapid antigen tests of the 18 tests that had an effective reproduction number ($R_E$) below one for testing frequencies ranging from every day to every 14 days and isolating positives. The whiskers denote the 95% credible interval based on 1,000 samples of an RT-PCR diagnostic sensitivity curve and rapid antigen percent positive agreement curve.

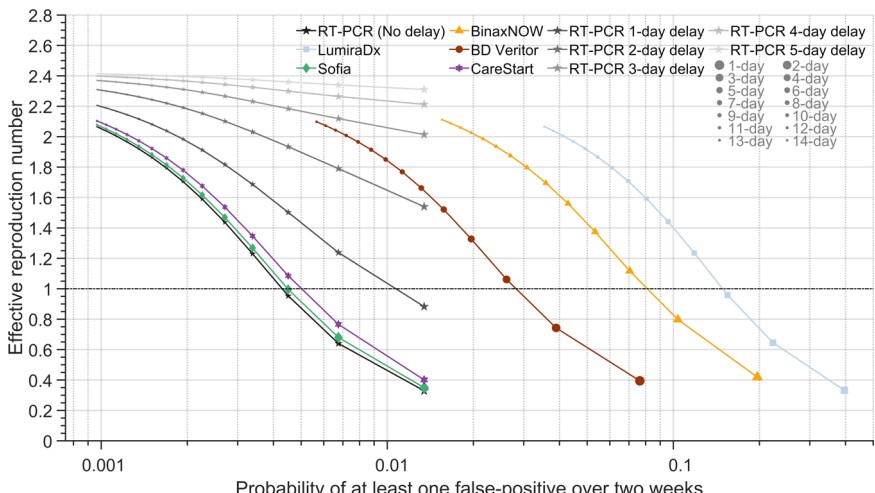

**Fig. 4 The effective reproduction number, and risk of one or more false positives, for varying frequencies of serial testing with RT-PCR and rapid antigen tests.** Specifying 35.1% of infections being asymptomatic, a 24-h delay before receiving RT-PCR results, no delay before receiving rapid antigen test results, an incubation period of 4.4 days, self-isolation upon symptom onset, and a RT-PCR diagnostic sensitivity curve informed by data from Hellewell et al.[11], the expected transmission with serial testing using RT-PCR test with no delay to five-day delay (black stars gradient) in obtaining test results and the rapid antigen test LumiraDx (light-blue squares); Sofia (green diamonds); BinaxNOW (yellow triangles); BD Veritor (dark-red circles); and CareStart (purple hexagrams) for testing everyday to every 14 days and isolating positives (small dots: longer time between tests; larger dots: shorter time between tests) and the corresponding probability of at least one false positive over a two-week period (x axis).

assured. This threshold exhibited only a moderate effect on the optimal testing strategies for some of the more effective RA tests. Specifically, the duration of quarantine needed to be extended by one day for four RA tests (CareStart (NS); Celltrion DiaTrust; LumiraDx (AS); Simoa) to achieve a probability of PQT equivalent or lower than a 7-day quarantine with an RT-PCR test on exit. In the context of serial testing, five RA tests (Clip COVID; Liaison (AS); Simoa; Sofia; Sofia 2 Flu + SARS) required the time interval between tests to be shortened by one day (i.e., more frequent testing; Supplementary Data 6). Other than these observations, our results remained consistent with those obtained in our baseline analysis (Supplementary Data 6).

To examine the effect of the proportion of infections that are asymptomatic ($p_A$) and the basic reproduction number ($R_0$) on quarantine duration and the frequency of serial testing, we conducted a two-way sensitivity analysis for the five commonly used RA tests. For varying values of $p_A$ and $R_0$, we determined the minimum quarantine duration that results in equivalent or lower probability of PQT than that computed for the RA test conducted on exit from a seven-day quarantine under the baseline parameterization ($p_A = 0.351$, $R_0 = 3.2$). The minimum quarantine required was positively associated with both the parameters (Supplementary Figs. 30a–34a). As $R_0$ decreases, the frequency of serial testing required to maintain $R_E < 1$ becomes increasingly sensitive to changes in $p_A$ (Supplementary Figs. 30b–34b). As $p_A$ increases for a specified $R_0$, we found that more frequent serial testing should be conducted to maintain $R_E < 1$ (Supplementary Figs. 30–34).

Compared to the RT-PCR diagnostic sensitivity curve (log Normal) applied in the baseline analysis, an alternative functional form (log Student's $t$) for the temporal sensitivity yields a higher probability of detecting infection over a longer duration, but a lower probability of detecting a case at the peak of infection (Supplementary Fig. 1). Under this alternative RT-PCR sensitivity curve, the results for the probability of PQT were largely similar to those in the baseline analysis (Fig. 2 vs Supplementary Fig. 35; Supplementary Datas 4 vs 7). The primary difference for the alternative RT-PCR curve was that CareStart (NS), Celltrion DiaTrust, Ellume, LumiraDx (AS), and Simoa required a one-day longer quarantine period for testing on entry and exit from quarantine than their corresponding quarantine duration in the baseline (Supplementary Data 4 vs 7). For serial testing, 10 of the 18 RA tests required testing every two days under the alternative RT-PCR curve instead of every three days to maintain $R_E$ below one (Supplementary Data 4 vs 7; Supplementary Figs. 24 vs 36; 25 vs 37).

**Comparing percent positive agreement data sets.** The PPA data used to inform the diagnostic sensitivity for the RA tests were obtained in a more clinical trial setting for companies seeking EUA from the U.S. Food and Drug Administration (i.e., internal) than when they may be applied in a real-world setting by independent investigators (i.e., external). Three of the RA tests (CareStart, Sofia, and BinaxNOW) had data available from studies evaluating their abilities to detect infections within a community. We conducted distinct analyses for these available internal and external datasets and compared the results. Internal datasets for both CareStart and Sofia were not significantly different from the external datasets, while they were significantly different for BinaxNOW (likelihood ratio tests; Supplementary Table 2).

Around the time of symptom onset, the diagnostic sensitivity informed by the internal dataset for Sofia and CareStart was greater than that inferred from the external dataset (Supplementary Figs. 20–22). In contrast, for BinaxNOW, the diagnostic

---

**Table 1 Paired BD Veritor and RT-PCR tests conducted, results, their agreement, and false positives.**

| Quarantine day after entry[a] | Day 0[b] | Day 3 | Day 4 | Total |
|---|---|---|---|---|
| Paired tests conducted | 818 | 726 | 675 | 2219 |
| RT-PCR tests positive | 12 | 5 | 3 | 20 |
| RA tests positive | 4 | 3 | 0 | 7 |
| RA test false negative [c] (RA−/RT-PCR +) | 9 | 4 | 3 | 16 |
| RA test false positives [c] (RA +/RT-PCR −) | 1 | 2 | 0 | 3 |
| RA test true positive [c] (RA +/RT-PCR +) | 3 | 1 | 0 | 4 |

[a]BHP onshore quarantine testing, 22 November 2020 to 17 January 2021.
[b]Test on entry into quarantine.
[c]Assessed by comparison to RT-PCR.

---

sensitivity estimated from the external dataset was always greater than that determined from the internal dataset (Supplementary Fig. 22). We found that the required quarantine duration was at most two days longer under the external dataset as compared to the internal dataset for Sofia, whereas there was no change for the BinaxNOW and CareStart tests. The frequency of testing was decreased by at most one day for BinaxNOW, increased by one day for Sofia, and no change for CareStart.

**Validation of serial RA testing within and subsequent to quarantine.** To examine the agreement between a RA test and RT-PCR in a cohort that was prescreened for COVID-19 symptoms, paired testing of BD Veritor and RT-PCR was conducted upon entry to quarantine, and on day three, and day four of a five-day quarantine (Table 1). From this paired testing within a prescreened cohort, the fraction of RA tests in agreement with negative RT-PCR was 2196/2199 (99.86%). This proportion was not significantly different from the fraction of RA tests in agreement with negative RT-PCR from the EUA documentation for BD Veritor (212/213 = 99.53%; two-tailed Fisher Exact Test $P = 0.309$). The PPA between the BD Veritor RA test and RT-PCR test declined over the course of quarantine, starting at 25% (Clopper-Pearson 95% CI:5.5–57.2%) on entry, to 20% (Clopper-Pearson 95% CI:0.5–71.6%) on day three, to 0% (Clopper-Pearson 95% CI: 0–70.8%) agreement by day four (Table 1). All 16 false negatives exhibited high cycle times (Ct ≥ 25), indicating an early or late phase of asymptomatic (or presymptomatic) disease and low levels of viral RNA, during which RA tests are less sensitive than RT-PCR (Supplementary Table 3). Three of the four true positive RA tests exhibited low cycle times (Ct ≤ 19) indicating high levels of viral RNA and (possibly) that testing was conducted mid-disease when RA tests have been identified as exhibiting higher sensitivity (Supplementary Table 3).

For the model-based analysis conducted in this section, an incubation period of 5.72 days is specified because the Delta variant was not yet dominant in the U.S.A. during the study period[58,59]. Specifying isolation on symptom onset and removal from quarantine after a positive RT-PCR test, the PPA between the BD Veritor with RT-PCR derived from our analysis was 28.5% (95% CrI: 20.3–32.7%) on entry, 28.3% (95% CrI: 20.8–32.1%) on day three, and 17.1% (95% CrI: 13.5–18.9%) on day four. The probability of obtaining 0% PPA for the three individuals that tested positive with RT-PCR on day four was 0.571 (95% CrI: 0.534–0.647).

The BD Veritor test was conducted offshore either on days three, six, and nine, or days two, five, and eight after a 24-h quarantine with an RT-PCR test on entry. A total of 1714 antigen tests were performed. A total of three RT-PCR confirmed cases

**Table 2 Serial offshore BD Veritor rapid antigen tests, positives, and false positives on days 3, 6, and 9.**

| Day after off-shore entry[a] | Day 3 | Day 6 | Day 9 | Total |
|---|---|---|---|---|
| RA tests conducted | 458 | 458 | 457 | 1373 |
| RA tests positive | 2 | 3 | 0 | 5 |
| RA false positives[b] | 2 | 0 | 0 | 2 |

[a]BP platform test data, 2 March 2021 to 22 May 2021.
[b]Subsequent onshore laboratory RT-PCR was negative.

**Table 3 Serial offshore BD Veritor rapid antigen tests, positives, and false positives on day 2, 5, and 8.**

| Day after offshore entry [a] | Day 2 | Day 5 | Day 8 | Total |
|---|---|---|---|---|
| RA tests conducted | 124 | 121 | 96 | 341 |
| RA tests positive | 0 | 0 | 0 | 0 |
| RA false positives | 0 | 0 | 0 | 0 |

[a]BP platform test data, 5 March 2021 to 24 May 2021.

were identified within the first six days of being offshore (Tables 2 and 3), with all individuals being asymptomatic at the time of RA positive results. Assuming that all detected cases did not develop symptoms, the maximum likelihood estimate of the number of asymptomatic infectious individuals in the offshore environment was 4 (95% CrI: 1–11), which corresponds to an effectiveness of the offshore testing strategy at identifying cases of 75% (95% CrI: 36.8–100%; Supplementary Fig. 38).

## Discussion

Our study demonstrates the utility of RT-PCR and RA tests in suppressing onward transmission of COVID-19 in the contexts of quarantine and serial testing. Examining 18 RA tests that have received EUA from the U.S.A. FDA, we showed that conducting RA tests on entry and exit can provide a greater reduction in post-quarantine transmission than conducting a single RT-PCR test 24 h prior to quarantine exit. RT-PCR results can be delayed by one or more days[60–62]. In that case, RA tests outperform RT-PCR at reducing onward transmission via serial testing and case isolation due to their faster turnaround time. Accordingly, the frequency of serial RT-PCR testing required to suppress outbreaks increases substantially with the associated waiting times in obtaining results. While RA tests can reduce $R_E$ below one with equivalent or less-frequent testing than the RT-PCR tests, the probabilities of false positives are markedly greater. With the use of RA and RT-PCR testing data from a remote industrial setting, we validated our estimates of the PPA with RT-PCR of the BD Veritor RA test during a 5-day quarantine. Based on a total of 1714 BD Veritor RA tests conducted between March 2, 2021 and May 24, 2021, it was determined that serial testing with this RA test every three days identified and isolated three of an estimated four asymptomatic infected individuals.

For quarantine policies with an exit test, RT-PCR tests typically must be conducted at least a day prior to the end of quarantine. In contrast, RA tests can be used closer to the end of quarantine due to rapid turnaround times. As a result of increasing test sensitivities during short quarantines[4], most RA tests conducted on exit outperformed RT-PCR tests for quarantine durations less than three days. Supplementing an exit RA test with one upon entry to quarantine could allow RA tests to outperform a single RT-PCR test on exit for longer quarantine durations. Specifically, for more than 50% of RA tests considered, conducting a RA test

on both entry and exit from quarantine produced a lower probability of post-quarantine transmission than an RT-PCR test on exit for quarantine durations up to 14 days. As prevalence in the community increases, having at least one false-negative in a cohort becomes more probable. Conducting a RA test on entry in addition to the one on exit mitigates the reduced sensitivity of a single test, thereby reducing the probability of releasing a still-infectious case from quarantine. Testing on alternative days to quarantine entry could increase the diagnostic sensitivity of the testing strategy even more than adding a test on entry. If logistically and financially feasible, diagnostic sensitivity can be increased further by testing on additional days. At a higher cost than a RA test, multiple RT-PCR tests could also be conducted in quarantine to increase the probability of identifying a case. Long quarantine durations have low practicality for entities trying to minimize interruptions to operations. By adding an additonal less costly RA test at entry or earlier than the exit test, the quarantine duration could be reduced, substantially mitigating loss of productivity attributed to quarantine. Therefore, RA tests can be suitable alternatives to a single RT-PCR, especially for short quarantines of one or two days.

RT-PCR tests exhibit greater diagnostic sensitivity than RA tests[35,63–65], so it may seem counterintuitive that the RA testing could provide equivalent performance or outperform RT-PCR testing during serial testing. This counterintuitive result is consistent with previous research[9] and arises from the ability of RA tests to identify infected individuals within minutes, leading to swift isolation of cases prior to substantial transmission. In contrast, obtaining results from RT-PCR sampling may be substantially delayed, taking hours or potentially days in high-prevalence or resource-limited settings. Utilizing rapid RT-PCR instrumentation would enable testing with shorter turnaround times[66]. However, major supply chain and maintenance hurdles for deploying large-scale point of care RT-PCR testing limit their broad applicability and remain infeasible for some remote settings. Delays of testing for surveillance purposes have a greater effect on transmission than delays of testing for quarantine, in turn requiring more frequent testing to prevent outbreaks (i.e., sustaining $R_E < 1$). More frequent testing would then give rise to logistical challenges and increased costs and could further slow RT-PCR turnaround times, resulting in a positive-feedback loop concerning delays and exacerbating negative outcomes. Our estimate argues for the efficacy of testing every two to three days with a RA test is consistent with empirical studies that have found this frequency of testing to provide an effective tool for infection surveillance within schools[67]. Therefore, rapid turnaround times for testing are essential to the utility of a surveillance program intended to suppress the spread of disease.

Using RA tests can mitigate some of the challenges and costs of serial testing with RT-PCR. However, RA tests can increase the chance of producing a false positive. From an operational standpoint, sending numerous or essential employees home could be critically problematic due to false-positive results[68]. Several factors drive the consequences of surveillance frequency in terms of the number of false positives and extent of transmission, including (i) the sensitivity and specificity of the test, (ii) the number of people tested, (iii) the frequency of testing, and (iv) the background prevalence of the disease. For example, in a low-prevalence setting the probability of obtaining at least one false-positive in a cohort increases geometrically with the number of people tested and similarly with the frequency of testing per individual. Less frequent RA tests will reduce the chances of a false-positive during serial testing in a low-prevalence community, but at the cost of an outbreak occurring upon missing an infectious individual ($R_E > 1$). Assiduous follow-up testing of identified potential cases that are isolated should be incorporated

—especially in remote offshore settings where false-positive cases may require extensive and costly measures such as evacuation. Given the higher specificity of RT-PCR, using an initial RA test with an RT-PCR follow-up for any positive can substantially reduce the number of RT-PCR tests, especially in a low-prevalence setting[69]. A challenge to follow-up with RT-PCR testing is not just the increased logistical and economic costs: false-negatives can result in reintroduction into the surveilled population and onward transmission. Multiple follow-up tests are likely necessary to investigate conflicting test results. Accordingly, there are tradeoffs between the risks of transmission, number of tests conducted, processing times, and underlying costs from false positives regardless of the test utilized in surveillance. Therefore, policy decision-makers can adapt the testing schemes based on the current status of the epidemic to a level of risk that they find acceptable.

The effectiveness of quarantine and serial testing strategies is also dependent on the effective reproduction number, which in turn is influenced by disease interventions. For example, a reduction in the effective reproduction number would occur during vaccine rollout when immunity increases in the population. We evaluated quarantine and serial testing strategies under the assumption that individuals self-isolate upon symptom onset, resulting in an $R_E$ of 2.4 when a basic reproduction number of 3.2 was specified. Organizations, institutions, and localities will likely implement multiple disease-control measures, resulting in idiosyncratic $R_E$. Higher $R_E$ or lower levels of self-isolation upon symptom onset will require more stringent testing strategies than indicated here.

Quantitatively, our results are dependent on epidemiological context. For example, breakthrough cases have been observed to less frequently be symptomatic and less frequently be subject to severe disease compared to unvaccinated cases[70,71]. As illustrated in our scenario analyses, the equivalent quarantine durations to RT-PCR and effective frequency of serial testing could increase due to asymptomatic infections despite a reduction in the reproduction number. Considering community prevalence instead of transmission, investment in a testing strategy with greater specificity might be a more prudent approach in low-prevalence settings, whereas a testing strategy with high sensitivity becomes more important in communities with high prevalence. Nevertheless, the heterogeneity of diagnostic sensitivity of tests that we have demonstrated is independent of the context-specific basic reproduction number. We expect the relative performance of RA tests in comparison to RT-PCR to remain intact despite manifest differences in $R_E$.

The accuracy of our results is dependent on the quality of data collected. Numerous factors influence the diagnostic sensitivity of RA tests relative to RT-PCR, such as the level of training of those who collect the samples, anatomical collection site, and storage conditions[72–75]. Substantial heterogeneity across RA tests exists in the assessment period of their sensitivity. For example, tests such as BinaxNOW, LumiraDx, and Simoa provided PPA data extending to at least 11 days post symptom onset. Other tests, such as Clip COVID, SCoV-2 Ag Detect, and Sofia 2 Flu + SARS provided results only extending five days beyond symptom onset (Supplementary Data 1). A limited number of samples at some time points can lead to uncertainty in the longitudinal diagnostic sensitivity of the RA tests. For example, the single data point at the last observed time for the Ellume RA test was the only one that did not agree with RT-PCR. This thresholded disagreement yielded a rapid decline in the logistic regression model for the PPA. Establishing a minimum sample size that is sufficient to provide statistical power for inference of sensitivity across time-points spanning the disease course would improve sensitivity

measurements and facilitate refinement in testing frequencies and necessary quarantines to suppress transmission.

All of the PPA data for RA tests was gathered subsequent to symptom onset. This absence could be concerning because the early sensitivity of RA tests is a crucial aspect of their utility. Inaccurate characterization of early-phase diagnostic sensitivity would affect the results for entry testing, quarantines shorter than the incubation period, and the effectiveness of frequent serial testing. However, we were able to address the absence of PPA data in the incubation period by constructing a mapping between the infectivity and the PPA.

Most PPA data was gathered within the week following symptom onset, with some RA test datasets extending to later time points. This dearth of data later in the time course leads to low precision in the extrapolation of the PPA beyond the extant data points, and high uncertainty regarding late-disease diagnostic sensitivity. The late-disease diagnostic sensitivity of the RA tests extrapolated beyond the data points should be applied with caution in other contexts where it might have consequences. However, the extrapolations and high uncertainty in RA-test late-disease diagnostic sensitivity has a limited effect on our findings (affecting estimates of quarantine duration and frequency of testing by at most a day), due to the low infectiousness and consequently limited transmission that occurs late in disease.

Because all results presented here relate to transmission, a higher probability that a RA test produces a false-negative result (compared with RT-PCR)[29,72] in the late stages of disease—when the individual may no longer be infectious—matters far less than their higher probability of producing a false-positive result during early stages of the disease. RA tests exhibit distinct PPA trajectories, especially late in disease; however, the estimated quarantine durations and frequencies of serial testing were relatively robust among the 18 RA tests (differentiating at most a day). Therefore, we expect little to no change in these estimates as additional information about the diagnostic sensitivity over the disease course for each RA test emerges.

The frequency of serial testing to maintain $R_E < 1$ was robust under alternative infectivity curves and incubation periods for the majority of the RA tests, with reductions in $R_E$ being consistent with previous studies[8,9,17]. However, the combined change in the infectivity profile and incubation period influenced the utility of RA tests in quarantine. When there is a low probability of identifying a case in the incubation period and more transmission after the incubation period, an RT-PCR on exit from quarantine reduces PQT more than a RA test on both entry and exit for quarantines of six days or longer duration. With the emergence of Omicron, the US CDC has shortened the recommended 7-day quarantine to a 5-day quarantine after exposure, followed by five days of strict mask use[76]. Omicron is estimated to have an incubation period of about three days[77,78], roughly two and a half days shorter than the estimated incubation period of the original pandemic virus (i.e., our alternative scenario). The infectivity profile of Omicron has not yet been determined. Preliminary evidence indicates that the viral load peaks between three to six days after diagnosis or symptom onset when infected with the Omicron variant[79,80]. This later peak in viral load relative to prior variants could indicate a shift toward a greater proportion of transmission occurring after symptom onset. Under these conditions, we would hypothesize that RA tests will be less effective at identifying cases during a 5-day quarantine compared to RT-PCR. Diagnostic sensitivity of the RA tests has remained fairly stable as SARS-CoV-2 has evolved[81–83], but could, in principle, change with new emerging variants[73]. Our scenario analyses suggests that epidemiological characteristics of the disease (e.g. incubation period, infectivity profile, proportion of asymptomatic

transmission) impact testing strategies more than moderate changes in diagnostic sensitivity.

The effectiveness of RA tests in a real-world setting in mitigating onward transmission is unclear. Improper sampling or analysis of the RA tests could influence the rates of false positives and false negatives[84,85]. Previous studies have found that the diagnostic sensitivity of self-administered RA tests was quantitatively lower—although not statistically different—from their diagnostic sensitivity when sample collection was conducted by a trained medical person[73–75,86]. Independently evaluated datasets on the testing of COVID-19 infections in the community were available for three of the 18 RA tests[32,49,50], enabling a comparison to the datasets provided from the EUA documentation. In two of the three comparisons (Sofia and CareStart), diagnostic sensitivity in independent analyses was lower than ideal conditions. The estimated quarantine duration and testing frequency were equivalent or more stringent under the internal dataset compared to the external dataset for two of the three tests (BinaxNOW and CareStart). Compared with outcomes for BD Veritor in quarantine and serial testing, we found that the estimates from our model were consistent with testing conducted among employees by medical personnel. These findings provide reassurance that internal datasets do not deviate substantially from the real-world utility and effectiveness of RA tests for disease control.

Further assurance of the real-world sensitivity of these tests (and therefore their effectiveness in the context of quarantine, serial testing, and isolation) awaits additional data reporting application of RA tests in additional community settings. Our field test data demonstrated that serial offshore RA testing can be deployed with prescreening and a 24-h onshore quarantine with an RT-PCR test on entry while still maintaining an acceptable level of risk. The prevalence of disease that is potentially departing to offshore platforms is reduced by prescreening individuals entering quarantine. The extent of prospective secondary infections is diminished further with the combined 24-h quarantine and RT-PCR testing. With RA testing being more effective for short quarantines, they could also be a suitable alternative for an entry test and immediate departure protocol. Our analysis for the BD Veritor RA tests highlights that testing every two days is sufficient to maintain $R_E < 1$ for a basic reproduction number of 3.2. The testing frequency was every three days on the offshore platform, although only conducted for part of the worker's time offshore. Upon identification of an RT-PCR confirmed positive test, contact tracing mitigated additional transmission chains. Therefore, the data from this remote, densely populated setting illustrates the effective integration of RA tests into the layered control measure approach for mitigating disease outbreaks.

Our analyses of 18 COVID-19 RA tests currently approved for use in the USA provide a comprehensive comparative assessment with reference to the gold-standard RT-PCR. Our results highlight the scenarios in which RA tests would serve as a suitable and even beneficial alternative to RT-PCR tests, incorporating the temporal dynamics that crucially describe their sensitivity and specificity, and characterize the operationally important outcomes: for quarantine, post-quarantine transmission, surveillance and isolation, and for suppression of the reproduction number. New tests should similarly be evaluated by these key measurable outcomes, so that their utility can be compared for strategy implementation. Such analyses provide important insights into the possible trade-offs in decision-making processes on the type of tests to use for both congregate and community settings.

## Data availability
All data generated, analyzed, and used to build the graphs and tables in this study are available in the online Zenodo repository (https://doi.org/10.5281/zenodo.6518442

(2022))[87] or presented in the published article, supplementary information, and supplementary datasets. The RT-PCR testing dataset used to infer the temporal RT-PCR diagnostic sensitivity is provided from the published article by Hellewell et al.[11]. The PPA datasets used in the inference of the temporal diagnostic sensitivity of each rapid antigen test are found in Supplementary Data 1. The specificity of each test is summarized in Supplementary Data 3. The cycle times for rapid antigen test false negatives and true positives for the paired testing of BD Veritor and RT-PCR are located in Supplementary Table 3. Source data for all graphs and other tables are available from the Zenodo repository[87]. All data are available from the corresponding author [J.P.T.] on reasonable request.

## Code availability
The computational code used to conduct the analyses and produce the figures and tables was implemented in MATLAB, and is available online[87].

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

## Acknowledgements

This study was supported by the Notsew Orm Sands Foundation, B.H.P., and B.P. S.M.M. acknowledges the support from the Canadian Institutes of Health Research [OV4 – 170643, COVID-19 Rapid Research] and the Natural Sciences and Engineering Research Council of Canada, Emerging Infectious Disease Modelling, MfPH grant.

## Author contributions

G.K., A.P.G., and J.P.T. conceived and designed the study with contributions from S.M.M., B.S., C.R.W. and A.P. C.R.W., A.P. and J.P.T. developed analytical approaches. C.R.W. wrote computational code with contributions from A.P. C.R.W. executed analyses, with guidance from A.P. and J.P.T. C.R.W., A.P., S.M.M., B.H.S., G.K., R.J.L.H., D.E.T., J.P.A.K.M.P., A.M.D., A.P.G., and J.P.T. contributed to interpretation of results. C.R.W. drafted the manuscript with contributions from A.P. and J.P.T. C.R.W., A.P., S.M.M., B.H.S., G.K., R.J.L.H., D.E.T., J.P.A.K.M.P., A.M.D., A.P.G., and J.P.T. contributed to the revision of the manuscript and approved the final version of the manuscript.

## Competing interests

The authors declare no competing interests.
