## [Peer Review File · Communications Medicine]

Reviewers' comments:

Reviewer #1 (Remarks to the Author):

This paper models the use of rapid antigen tests to make decisions on the length of quarantine for individuals infected with SARS-COV-2. The specific motivation is industrial settings such as offshore platforms. The main contribution is to perform this modeling using sensitivity-over-time curves for a number of antigen tests currently available. Some new primary data is also given about agreement between antigen tests and PCR tests.

Providing specific and empirically grounded recommendations about testing strategies to determine when individuals should stay in quarantine is a worthwhile contribution. The high-level conclusions and methods employed appear reasonable overall. I have few main concerns, all addressable in a revision:

(1) The modeling treats the accuracy of antigen tests as their agreement with a PCR test. However, this is not necessarily correct, particularly during the period of recovery from infection. In this later stage, it is likely that individuals who test RA-negative but PCR-positive are no longer highly infectious (a point the authors allude to with a mention of high CT values for such cases). This is likely consequential in evaluating the utility of RA tests in ending quarantines.

(2) The organization of the manuscript as a whole is a little confusing. In particular, the new data presented about agreement between PCR and RA tests is not really integrated into the rest of the paper (the modeling uses only data from previous studies). It's not entirely clear what the purpose of including this data in this paper is, or what new lessons are to be drawn from it. It's not necessarily a bad thing to present some more evidence confirming what existing studies have shown about RA tests, but it would be helpful for their to be more explicit positioning of what we're supposed to take away from this section.

(3) The results are based on a logistic model for agreement between the RA and PCR tests over time. However, there are plenty of cases in the supplement where the logistic model does not appear to fit well, or where the model extrapolates very far past the point where any data is available at all. What is the motivation for specifying a logistic model? How can we be confident that it provides reliable extrapolations? One point of misspecification appears to be that RA tests have lower sensitivity in the very early stages of infection, which isn't captured by having a single (sign-constrained) coefficient. This might cause some issues with estimation, where the fit is a flat line (over time) because the model can't capture non-monotonic behavior.

I also have some more minor comments:

(1) The basic reproduction number assumed (2.5) seems a little out-of-date, given that delta is likely to be the predominant strain. The authors note that their qualitative conclusions will not be impacted by changes to R_0 , which is likely true, but the recommendations about quarantine lengths certainly will be. A sensitivity analysis of some of the key results to a higher value of R_0 would improve the paper.

(2) What role is being served by the RA test on entry to quarantine, as opposed to a test only at the end? This wasn't immediately clear.

(3) What is the rationale for using the incubation period distribution provided by Qin et al instead of Lauer et al?

Reviewer #2 (Remarks to the Author):

This manuscript examines testing, isolation (of cases) and quarantine (of non-cases and non-symptomatic individuals) strategies in order to reduce the risk of SARS CoV-2 transmission, primarily within occupational settings. The authors use a model of infectivity averted, translated to an effective R measure, to assess PCR (assumed 1 day turnaround) and 18 Rapid Antigen tests (no turnaround time) in 1) quarantine and testing strategies, 2) repeat testing strategies, and 3) to examine expected numbers of false positive from different testing frequencies and technologies. The paper uses data on the percent positive agreement post symptom onset with PCR for a range of RA tests in order to obtain test sensitivity curves, using data from manufacturers but also presenting data from empirical screening studies (for the more limited number of tests for which this was available). PCR positivity is estimated with data from Hellewell et al, with some modifications to their method. A second study inferring PCR positivity is used as a sensitivity analysis. The study found that the transmission equivalent of a 7-day quarantine with PCR on exit policy could be broadly achieved using RA tests, with some variability among them (required quarantines of 6-11 days across the 18 tests). Assuming a 24 hr turnaround delay for diagnosing and isolating cases for PCR tests, the study found serial testing with tests taken every 6 days to be sufficient to keep $R < 1$ for PCR, and every 6-8 days for RA tests, with long PCR test return delays shifting favour towards RA tests. However, more intensive testing strategies using RAs also led to more false positives.

Overall, the approach of the work is sound and the qualitative conclusions, that lack of delay in obtaining RA test results as compared to PCR could at least or more than make up for lower overall sensitivity of the tests, are in line with the findings of other studies (eg Larremore et al <https://www.ncbi.nlm.nih.gov/pmc/articles/PMC7325181/>). The other paper I am aware of examining multiple RA tests (Pickering et al) does not go on to consider the implications for transmission within testing interventions. This paper could therefore constitute a helpful addition to the literature on choice of testing technology. It is useful that the paper also present empirical data on a test screening intervention for offshore workers, as well as modelling analyses. However, the use of a longer incubation period estimate than that seen in many other studies potentially is puzzling (mean 8.3 days compared to 5-6 days) and has implications for the findings because I think it affects the analysis of RA test sensitivity relative to PCR and the infectivity profile. I think it might lead to longer intervals between tests being found adequate to keep $R < 1$ (along with a low estimate of R_0). The estimate used is from a single study from very early in the epidemic and I cannot see a justification or rationale given for this choice compared to the lower estimates found in systematic review studies. I think it would therefore be helpful for the authors to give a strong justification for why this estimate is used and to conduct sensitivity analyses as to the effect on their findings. The authors should be commended for using a second estimate of the PCR test sensitivity curve, and this looks more like what I would expect for PCR relative to RA test sensitivity. My other concern is that, as I understand it, there is relatively little heterogeneity in infection progression incorporated across individuals (eg in the form of incubation periods or infectivity profiles resulting) and the findings are not expressed with any uncertainty or variability.

Major points

1. It was not clear to me why the authors preferred an incubation period of 8.29 days, from Qin et al. This estimate, from very early in the pandemic, is higher than that of many other studies, typically 5-6 days, and it is higher than the mean incubation period found in systematic reviews. I think if the authors choose to use this longer estimate it needs to be strongly justified given its importance to the analysis. Ideally they could conduct some sensitivity analyses to assess the extent to which findings might vary given assumptions about the distribution of the incubation period. I have not formally reviewed estimates of the incubation period, but other systematic reviews seem to put it significantly lower than what the authors use, eg:

<https://www.acpjournals.org/doi/10.7326/m20-0504> 5.1 days (95% CI, 4.5 to 5.8 days) (also early 2020 data)

<https://systematicreviewsjournal.biomedcentral.com/articles/10.1186/s13643-021-01648-y> The mean incubation period ranged from 5.2 (95% CI 4.4 to 5.9) to 6.65 days (95% CI 6.0 to 7.2) (up to dec 2020)

<https://bmcinfectdis.biomedcentral.com/articles/10.1186/s12879-021-05950-x> the weighted pooled mean incubation period of COVID-19 was 6.5 (95%CI: 5.9–7.1) days (up to July 2020)

<https://bmjopen.bmj.com/content/10/8/e039652> (5.8 (95% CI 5.0 to 6.7) days) (up to April 2020)

As I understand it from the supplementary material, the longer assumed incubation time distribution (Qin et al) will also be reflected in the estimate of the infectivity profile because the authors state that they use the Qin et al incubation period distribution in their estimation of the diagnostic sensitivity curve of the RT-PCR test from the Hellewell et al data. They then state that this diagnostic sensitivity is constrained to peak at the same time as infectivity over the course of infectious age (which does make sense assuming both are linked to viral load in the naso-pharyngeal tract). Could the authors clarify whether this means that the infectivity profile that they then obtain tends to also peak relatively late, compared to other studies, and to give a generation interval that is relatively longer on average? This would presumably mean that longer intervals between tests in a serial testing strategy would be found to be more effective?. Could the authors comment on how their findings about the frequency of tests required to maintain $R < 1$ compare to other studies, eg Larremore et al 2021 (although I appreciate it is difficult to make different studies directly comparable with different assumptions)?

I think that the long estimate for the incubation period also affects the extent to which RA tests are assumed to agree with PCR post-symptom since time since infection (Fig 1), because the only measurements for agreement were relative to symptom onset, which in this study is placed relatively late in infection. I think that if the incubation period was assumed to have a mean of approximately 5 days, the decline of RA test sensitivity post symptom onset would have occurred earlier in the infectious age, and so is also responsible for the better looking RA sensitivity relative to PCR in the later stages of infection, than I would expect. My expectations are based on analyses which estimate RA test sensitivity stratified by CT value/viral load (eg as in https://www.liverpool.ac.uk/media/livacuk/research/Mass_testing_evaluation.pdf). Because PCR is more sensitive than RA tests at lower viral loads, as the authors themselves do mention, I would have expected a slightly later rise and faster decline in RA test sensitivity relative to PCR than shown in Figure 1.

The test sensitivity curve obtained in the sensitivity analysis using the Wells et al data, Figure S21,

looks more like what I would expect for RA compared to RT-PCT sensitivity. The authors had modified the original Hellewell analysis and used the long Qin et al incubation period in the modified analysis - was this not also the case for the Wells curve, where the peak in PCR test sensitivity occurs earlier and is more consistent with a viral load, and therefore infectivity profile that peaks earlier than that assumed in the main paper analysis?

2. There is no variation or uncertainty expressed for the estimated testing and quarantining strategies, whereas in practice we would expect heterogeneity in the progression of infections (incubation period, infectious periods, duration of infectiousness, and therefore test sensitivity over infection) for different individuals. Could the authors explain why no variability is shown?

3. From the description of the available data on the % positive agreement with PCR of each RA test, it sounds like there was no data to inform the relationship between PCR and RA test sensitivity pre symptom onset and that this was inferred based on a mapping between the relative infectivity function and the percent positive agreement post symptom onset. The uncertainty over this period is understandable given the lack of data available, but should be stated as a limitation in the Discussion (especially because, rightly, test sensitivity early in infection is highlighted to be important).

4. The method used to obtain measures of RA test specificity (shown in table S4) seems to mean that RA tests can only have a specificity as good as, or lower than that assumed for RT-PCR, 99.90%. What is the justification for this? Could the authors comment on how their findings regarding RA test specificity compare with others, eg Public Health England (now UK Health Security Agency) estimated the specificity of the Innova RA test to be 99.97%, which is higher than that assumed for other RA tests in this study (<https://www.gov.uk/government/publications/lateral-flow-device-specificity-in-phase-4-post-marketing-surveillance/lateral-flow-device-specificity-in-phase-4-post-marketing-surveillance>).

4. It was a strength of the paper that the authors used a second study to determine the PCR sensitivity curve (well et al) over time since infection and compared their findings.

5. It is also a great strength of this study that empirical analyses of relative test performance, and of serial testing, are incorporated.

6. The value for R_0 of 2.5 is quite out of date for the Delta variant (or the previously dominant Alpha variant). I am very sympathetic to the fast-evolving epidemiological situation and what this means for keeping an analysis up-to-date, but I think the authors should at least discuss the implications of an R_0 higher than 2.5 on their findings.

7. Similarly, it would be helpful for the authors to explicitly reflect in the Discussion on how their findings translate to a setting with vaccination and past infection.

8. For the analysis of how testing might shorten the required quarantine period, the authors are not clear about what the quarantine period they discuss is for, ie whom they expect to be quarantining, for what reason. This is important because it has implications for the likely infectious age of infected individuals as they enter quarantine, and this affects both test sensitivity and transmissibility. This needs to be clarified and differentiated between the different scenarios throughout the paper. For instance for a close contact of a case entering quarantine, their likely infectious age will be a

function of the time between exposure to the case and the time that they are notified/traced. Whereas for offshore workers, infected individuals will presumably be random across their pre-symptomatic infectious age, (depending on symptom status and whether or not symptomatic individuals would be expected to already be quarantining). If the modelling is primarily to assist with the example of sending staff offshore, and when the randomly sampling of infectious age is more reasonable, this should be explicitly stated, rather than only mentioned as an example. (Though I would argue that because symptomatic individuals should be opting out anyway, you will tend to have more people earlier in their infections at the start of quarantine, than at the end).

9. Figure 1 shows the sensitivity of RT-PCR (and some of the RA tests, appearing to be 1 or very close to 1 at its peak. Is this correct? This is different to the original Hellewell et al paper in which PCR sensitivity peaks at approx. 80%. Or is the y axis scale intended to be relative to PCR at its peak? If the latter, please indicate this more clearly in the figure labelling and/or notes.

10. It is very good that the authors compared test sensitivity and specificity for both the company datasets and real-world data because one might expect these to differ. It would be useful when describing this on p10 to briefly describe who the external populations were and how they might vary or not compared to the target population (eg were they lay people self-swabbing, or swabbing conducted by a health professional?).

Minor points:

11. Abstract:

*Suggest changing COVID-19 infections in first sentence to SARS-CoV-2 infections.

*Suggest second sentence saying what is being compared (potential transmissibility?)

12. Introduction first paragraph : “For instance, previous studies have shown that a 14-day quarantine with no testing can safely be shortened to a seven-day quarantine if a nasopharyngeal RT-PCR test is conducted on exit from the quarantine 4–6, a reduction notified for widespread implementation by the Centers for Disease Control and Prevention (CDC)⁷.” : suggesting specifying that this is quarantine of a close contact, rather than isolation of a case (this distinction might not be clear to all readers).

13. p2, final paragraph - it would be worth mentioning too that the proportion of false positives will vary with prevalence so the extent to which we are worried about false positives versus false negative will change over the course the epidemic. Also of concern could be the behavioural effects of receiving a negative test (in case these might lead to a higher level of risk-taking than would have occurred with no test at all).

14. p3 :”One of the critical control strategies identified early on was systematic quarantine and testing” -> could the authors clarify the circumstances of the quarantine they are investigating? Is this a quarantine prior to joining the workforce offshore? Or is this a quarantine if a close contact of someone who has tested positive in an offshore environment (or any environment)? As mentioned above, there are implications for where in their infectious age we might expect any quarantined cases’ infectious ages to be.

15. P4: could the final paragraph of the introduction clarify that the analysis was specifically

designed to inform quarantine and serial testing strategies for offshore workers in the extractive industry - this is implied but not so clearly stated.

16. "With a negligible delay in turnaround time, RT-PCR tests can be performed just once every eight days." : please could a more specific figure be given instead of 'negligible'. Does this mean <24 hrs, 12 hrs, no delay?

17. Figure 2 caption: I think the RT-PCR test on exit line line was a solid black line with stars, not circles.

18. It makes sense to me that for short quarantines, RT-PCR test on exit strategies would be less effective than the RA tests as modelled because they are taken a day earlier than the RA tests, and the probability of testing positive at this point in time is lower than RA tests taken a day later? This then switches around 6 days of quarantine, after which the effect of the higher sensitivity of PCR later on in infection dominates - is that the intuition for why this finding is observed? (Fig 2A). And then in Fig 2B, the relative lack of sensitivity of a single RA test result later in infection is made up for by the improvement in sensitivity via having taken an RA test on entry as well? It might be worth highlighting this intuition if so.

19. Discussion : p14, about false-positives "and consideration should be devoted to appropriate follow-up testing of potential cases that are identified and isolated." -> I think this should be modified to account for community prevalence at the time of testing - our relative concern about false positives might vary accordingly (although it might be greater in the example of offshore testing na potential need for evacuation, for instance, where the cost is higher).

It is unclear to me how and whether the risk of a false positive from PCR was assessed? CHECK this for previous analyses.

20. Abstract: please specify the FDA, 'in the United States', to make the setting clear.

Reviewers' comments:

Reviewer #1 (Remarks to the Author):

This paper models the use of rapid antigen tests to make decisions on the length of quarantine for individuals infected with SARS-COV-2. The specific motivation is industrial settings such as offshore platforms. The main contribution is to perform this modeling using sensitivity-over-time curves for a number of antigen tests currently available. Some new primary data is also given about agreement between antigen tests and PCR tests.

Providing specific and empirically grounded recommendations about testing strategies to determine when individuals should stay in quarantine is a worthwhile contribution. The high-level conclusions and methods employed appear reasonable overall. I have few main concerns, all addressable in a revision:

(1) The modeling treats the accuracy of antigen tests as their agreement with a PCR test. However, this is not necessarily correct, particularly during the period of recovery from infection. In this later stage, it is likely that individuals who test RA-negative but PCR-positive are no longer highly infectious (a point the authors allude to with a mention of high CT values for such cases). This is likely consequential in evaluating the utility of RA tests in ending quarantines.

Response: We thank the reviewer for raising this critical difference between the PCR test and RA test. Indeed, evidence indicates that SARS CoV-2 cannot be successfully cultured beyond day nine of illness, yet PCR can still yield positive results. To address whether differences in the results of these tests among those late in disease and no longer infectious make a difference to our findings, we have conducted a scenario analysis where a hard cut-off is imposed such that the RA test after the cut-off will only return a negative test result. We found that in this case, with a 'hard' cut-off for non-zero sensitivity of the RA test in the later stages of disease, the impact on our quantification is marginal. The impact on transmission is marginal as a consequence of the very limited amount of transmission that occurs 10 days after symptom onset. Transmission at this late stage accounts for ~0.03% of all infection, by our accounting, consistent with our earlier published research ¹ and that of others ^{2,3}.

Furthermore, to ensure that our results are not sensitive to the late-stage difference in positivity comparing RT-PCR and antigen tests, we considered an even more extreme threshold of 5.6 days post-symptom onset, selected because it is just outside the minimum recommended window for use of the RA test ⁴. The more extreme threshold was specified based on the time post-symptom onset in which 1% of infection remains. Specifying this more extreme threshold, we found that the more effective RA tests (listed in text) required a day longer quarantine and shortening the time between serial testing by a day. We note that the threshold determines the minimum level of infectivity at which

¹ Wells, C.R., Townsend, J.P., Pandey, A. et al. Optimal COVID-19 quarantine and testing strategies. *Nat Commun* 12, 356 (2021). <https://doi.org/10.1038/s41467-020-20742-8>

² Ashcroft et al. *eLife* (2021) Quantifying the impact of quarantine duration on COVID-19 transmission. doi: 10.7554/eLife.63704

³ Quilty, Billy J., et al. "Quarantine and testing strategies in contact tracing for SARS-CoV-2: a modelling study." *The Lancet Public Health* 6.3 (2021): e175-e183. [https://doi.org/10.1016/S2468-2667\(20\)30308-X](https://doi.org/10.1016/S2468-2667(20)30308-X)

⁴ In Vitro Diagnostics EUAs - Antigen Diagnostic Tests for SARS-CoV-2 (Accessed Jan 1, 2022) USA FDA.

<https://www.fda.gov/medical-devices/coronavirus-disease-2019-covid-19-emergency-use-authorizations-medical-devices/in-vitro-diagnostics-euas-antigen-diagnostic-tests-sars-cov-2>

the RA test could return a positive result (i.e., the rapid antigen test returns only a negative value when the infectivity is less than the infectivity at the specified threshold).

Lastly, we have now made sure in the discussion to point out that an individual may be RA-negative but PCR-positive in the late stages of disease.

Added text to the Methods:

We conducted scenario analyses to determine the impact of (i) the incubation period; (ii) reduced diagnostic sensitivity of RA tests for low-levels of infectivity; (iii) proportion of asymptomatic infections and the basic reproduction number; and (iv) the RT-PCR diagnostic sensitivity curve on the onward transmission after quarantine and during serial testing.

To evaluate the robustness of our results to a potentially longer duration of incubation than 4.4 days 41, we evaluated the impact of an alternate incubation period of 5.72 days and the infectivity profile associated with that longer duration 37. A positive RA test indicates active infection, suggesting that there is a substantial concentration of virus within the sample site to return a positive. Studies have indicated that SARS-CoV-2 cannot be successfully cultured 10 days after symptom onset 44. Thus, it is possible at some stages of the disease that a RA test will only return a negative while RT-PCR still yields a positive. There is also uncertainty in the inferred diagnostic sensitivity in the later stages of disease, as the PPA was extrapolated past the last recorded data point for all RA tests. To address whether differences in the results of these tests among low levels of infectivity make a difference to our findings, we imposed a threshold on the level of infectivity such that the RA test will only return a negative. If the infectivity at the time of the RA tests is below the infectivity at 10 days post-symptom onset then the RA test returns a negative result (Fig. S42). We examined a stricter threshold based on the infectivity at 5.6 days post-symptom onset (where there is 1% infectivity remaining). To assess the sensitivity of our results to variation in the proportion of asymptomatic infection and the basic reproduction number, we conducted a grid analysis with values ranging from 0.1–0.95 and 1.05–5, respectively. Furthermore, we evaluated an alternative model for the temporal RT-PCR diagnostic sensitivity curve, substituting a log-Student's-t distribution for the log-Normal distribution used in the baseline analysis.

Added text to the Results:

We examined a scenario in which if the level of infectivity at the time of testing was less than the infectivity 10 days after symptom onset then the RA test only returns a negative result. For this specified threshold in RA test diagnostic sensitivity, we observed no change in the results (Table S6). To explore the impact of a narrower period in which a RA test can return a positive result, we imposed a higher threshold (i.e., the level of infectivity on 5.6 days after symptom onset). This higher threshold exhibited a moderate effect on the optimal testing strategies for some of the more effective RA tests. Specifically, the duration of quarantine needed to be extended by one day for four RA tests (CareStart (NS); Celltrion DiaTrust; LumiraDX (AS); Simoa) to achieve a probability of PQT equivalent or lower than a 7-day quarantine with an RT-PCR test on exit. In the context of serial testing, five RA tests (Clip COVID; Liaison (AS); Samoa; Sofia; Sofia 2 Flu + SARS) required a shorter time interval between tests by one day (i.e., more frequent testing; Table S6). Other than these observations, our results remained consistent with those obtained in our baseline analysis (Table S6).

Added text to the Discussion:

The concentration of the PPA data for most of the RA tests falls within the week following symptom onset. However, we addressed the absence of PPA data in the incubation period by constructing a mapping between the infectivity and the PPA. Moreover, due to low infectiousness and limited transmission in the late stage of disease, the higher uncertainty regarding the late sensitivity of RA tests only has a limited effect on our findings (affecting estimates of quarantine duration and frequency of testing by at most a day). For the results presented here, the higher probability that a RA test produces a false-negative result (compared with RT-PCR) 29,71 in the late stages of disease—when the individual may no longer be infectious—matters far less than that during the early stage of the disease.

Discrepancies with our inferences regarding early-phase diagnostic sensitivity could affect the results for entry testing, quarantines shorter than the incubation period, and the effectiveness of frequent serial testing. Even with the RA tests having distinct PPA trajectories from other RA tests, the estimated quarantine duration and frequency of serial testing were relatively robust among the 18 RA tests (differentiating at most a day). Therefore, we expect little to no change in our estimates as additional information about the diagnostic sensitivity over the disease course for each RA test emerges.

Added Table and Figure to the Supplementary material:

Table S6. The change in the required quarantine durations and serial testing frequencies when there is a specified threshold in which the rapid antigen test can return a positive test result relative to the results in Table S1 for the maximum likelihood estimates.

Rapid antigen test	Threshold based on infectivity at 5.6 days after symptom onset			Threshold based on infectivity at 10 days after symptom onset		
	Exit test ^a	Entry and exit test ^a	Frequency ^b	Exit test ^a	Entry and exit test ^a	Frequency ^b
BD Veritor ^{d,e}	0	0	0	0	0	0
BinaxNOW ^{e,f}	0	0	0	0	0	0
BinaxNOW ^{g,f}	0	0	0	0	0	0
BinaxNOW ^{h,f}	0	0	0	0	0	0
CareStart ^{e,f}	0	0	0	0	0	0
CareStart ^{e,i}	0	1	0	0	0	0
CareStart ^{f,g}	0	0	0	0	0	0
CareStart ^{f,h}	0	0	0	0	0	0
Celltrion DiaTrust ^{e,i}	0	1	0	0	0	0
Clip COVID ^{e,f}	0	0	- 1	0	0	0
Ellume ^{e,j}	0	0	0	0	0	0
Liaison ^{e,f}	0	0	- 1	0	0	0
Liaison ^{e,i}	0	0	0	0	0	0
LumiraDX ^{e,f}	0	1	0	0	0	0
LumiraDX ^{e,i}	0	0	0	0	0	0

Omnia ^{e,f}	0	0	0	0	0	0
SCoV-2 Ag Detect ^{e,f}	0	0	0	0	0	0
Simoa ^{e,i}	0	1	- 1	0	0	0
Sofia ^{e,f}	0	0	- 1	0	0	0
Sofia ^{h,f}	0	0	0	0	0	0
Sofia ^{g,f}	0	0	0	0	0	0
Sofia 2 Flu+SARS ^{e,f}	0	0	- 1	0	0	0
Status COVID-19/Flu ^{e,i}	0	0	0	0	0	0
Vitros ^{e,i}	0	0	0	0	0	0

^a Quarantine durations that are equivalent or better than a 7-day quarantine with an RT-PCR test conducted 24 h before exit.

^b The minimum required testing frequency for serial testing such that the effective reproductive number is less than one.

^c The probability of at least one false positive in a two-week period of serial testing under the minimum required testing frequency.

^d Peer-reviewed

^e Data from EUA submission

^f Anterior nasal swab

^g Data from community testing

^h Combined data from EUA submission and community testing

ⁱ Nasopharyngeal swab

^j Mid-turbinate swab

Figure S42. Temporal diagnostic sensitivity of a rapid antigen test when specifying a threshold day for returning a positive test. Specifying an incubation period of 4.4 days, the temporal diagnostic sensitivity of an RT-PCR test (black line), the BD Veritor rapid antigen test with no cut-off (red dashed line), the BD Veritor rapid antigen test with a threshold day specified by the level of infectivity 10 days after symptom onset (solid red line), and the BD Veritor rapid antigen test with a cut-off specified by the level of infectivity 5.6 days after symptom onset (solid yellow line).

(2) The organization of the manuscript as a whole is a little confusing. In particular, the new data presented about agreement between PCR and RA tests is not really integrated into the rest of the paper (the modeling uses only data from previous studies). It's not entirely clear what the purpose of including this data in this paper is, or what new lessons are to be drawn from it. It's not necessarily a bad thing to present some more evidence confirming what existing studies have shown about RA tests, but it would be helpful for their to be more explicit positioning of what we're supposed to take away from this section.

Response: We thank the reviewer for their feedback on improving the readability of the manuscript. We have restructured the Introduction, Results, and Discussion to improve the readability and flow, while expanding on the utility of new data, integrating it with our analytical framework, and what lessons can be drawn from it.

Revised text in Introduction:

Therefore, we present serial testing data for an offshore oil site as there is currently minimal published evidence of the effectiveness of large-scale serial RA testing in mitigating outbreaks within an industrial setting.

Revised text in Introduction:

Here we construct the temporal diagnostic sensitivity curves for 18 RA tests that have received EUA using data on percent positive agreement (PPA) with an RT-PCR test and temporal diagnostic sensitivity of an RT-PCR test. To determine when these RA tests can serve as a suitable alternative to the more costly and laborious RT-PCR tests, we calculated (i) their associated probabilities of post-quarantine

transmission (PQT) for quarantine durations from one to 14 days with testing on exit or both entry and exit upon random entry into quarantine; (ii) their extents of onward transmission for serial testing conducted every day to every 14 days; and (iii) their associated probabilities of false-positives during serial testing. We further evaluated the utility of RA tests using data collected from two offshore oil companies in the context of quarantine and serial testing within an industrial environment.

Added Text to the Methods:

Estimating the positive agreement between BD Veritor and RT-PCR during quarantine

We calculated the probability that an infected individual would test positive with both the RA test and RT-PCR on entry, day three of quarantine, and day four. This calculation accounts for isolation both upon symptom onset and upon a positive RT-PCR test.

Effectiveness of the offshore serial rapid antigen testing

To determine the number of infectious individuals that were in the offshore environment, we used our model framework for serial testing, the estimates for the specificity and sensitivity of BD Veritor from the FDA reports, the number of false positives and false negatives from the RA test, RT-PCR confirmed positives, and the probability of onward transmission from these cases using a maximum-likelihood approach.

Added Text to the Results:

For the model-based analysis conducted in this section, an incubation period of 5.72 days is specified because the Delta variant was not yet dominant in the United States during the study period^{58,59}. Specifying isolation on symptom onset and removal from quarantine after a positive RT-PCR test, the PPA between the BD Veritor with RT-PCR derived from our analysis was 28.5% (95% CrI: 20.2%–32.0%) on entry, 28.3% (95% CrI: 20.5%–31.5%) on day three, and 17.1% (95% CrI: 13.4%–18.5%) on day four. The probability of obtaining 0% PPA for the three individuals that tested positive with RT-PCR on day four was 0.571 (95% CrI: 0.541–0.650).

Added Text to the Results:

The BD Veritor test was conducted offshore either on days three, six, and nine, or days two, five, and eight after a 24-h quarantine with an RT-PCR test on entry. A total of 1714 antigen tests were performed. A total of three RT-PCR confirmed cases were identified within the first six days of being offshore (**Tables 2–3**), with all individuals being asymptomatic at the time of RA positive results. Assuming that all detected cases did not develop symptoms, the maximum likelihood estimate was 4 (95% CrI: 1–11) asymptomatic infectious individuals in the offshore environment, resulting in 75% (95% CrI: 36.8%–100%) effectiveness of the offshore serial testing strategy (**Fig. S36**).

Added text to the Discussion:

With the use of RA and RT-PCR testing data from a remote industrial setting, we validated our estimates of the PPA with RT-PCR of the BD Veritor RA test during a 5-day quarantine. Serial testing with BD Veritor in an offshore environment was estimated to identify three of four asymptomatic infected individuals.

Revised text in Discussion:

The effectiveness of RA tests in a real-world setting in mitigating onward transmission is unclear. Improper sampling or analysis of the RA tests could influence the rates of false positives and false negatives^{82,83}. Previous studies have found that the diagnostic sensitivity of self-administered RA tests was quantitatively lower—although not statistically different—from their diagnostic sensitivity when sample collection was conducted by a trained medical person^{73,74,84}. Independently evaluated datasets on the testing of COVID-19 infections in the community were available for three of the 18 RA tests^{32,49,50}, enabling a comparison to the datasets provided from the EUA documentation. In two of the three comparisons (Sofia and CareStart), diagnostic sensitivity in independent analyses was lower than ideal conditions. The estimated quarantine duration and testing frequency were equivalent or more stringent

under the internal dataset compared to the external dataset for two of the three tests (BinaxNOW and CareStart). Compared with outcomes for BD Veritor in quarantine and serial testing, we found that the estimates from our model were consistent with testing conducted among employees by medical personnel. These findings provide reassurance that internal datasets do not deviate substantially from the real-world utility and effectiveness of RA tests for disease control.

Added Text to the Supplementary Material:

Estimation of rapid antigen percent positive agreement with RT-PCR during quarantine.

To compare the results obtained from the paired testing of BD Veritor and an RT-PCR testing in a pre-screened cohort with that of our analytical framework, we computed the probability that both BD Veritor and RT-PCR test produce a positive on entry to quarantine, day three of quarantine, and day four.

Specifying random entry into quarantine in the absence of symptoms, the diagnostic sensitivity of the rapid antigen test $s_A(t)$, and the diagnostic sensitivity of RT-PCR $s(t)$, the probability that both BD Veritor and RT-PCR test produce a positive on entry to quarantine for a soon-to-be symptomatic is

$$Q_{S,0} = \frac{1}{t_s} \int_{u=0}^{t_s} s_A(u) \cdot s(u) du, \quad (S19)$$

and for an individual never exhibiting symptoms,

$$Q_{A,0} = \frac{1}{t_e} \int_{u=0}^{t_e} s_A(u) \cdot s(u) du. \quad (S20)$$

Individuals that test positive with RT-PCR—irrespective of the result from the rapid antigen test—or exhibit symptoms within quarantine are removed from the quarantine environment and placed in isolation. For a soon-to-be symptomatic individual, the probability that both BD Veritor and RT-PCR test produce a positive on day three of quarantine is

$$Q_{S,3} = \frac{1}{C_{S,0}} \int_{u=0}^{t_s-3} (1 - s(u)) \cdot s_A(u + 3) \cdot s(u + 3) du, \quad (S21)$$

where

$$C_{S,0} = \int_{u=0}^{t_s-3} (1 - s(u)) du. \quad (S22)$$

For an asymptomatic individual, the probability that both BD Veritor and RT-PCR test produce a positive on day three of quarantine is

$$Q_{A,3} = \frac{1}{C_{A,0}} \int_{u=0}^{t_e} (1 - s(u)) \cdot s_A(u + 3) \cdot s(u + 3) du, \quad (S23)$$

where

$$C_{A,0} = \int_{u=0}^{t_e} (1 - s(u)) du. \quad (S24)$$

For a soon-to-be symptomatic, the probability that both BD Veritor and RT-PCR test produce a positive on day four of quarantine is

$$Q_{S,4} = \frac{1}{C_{S,3}} \int_{u=0}^{t_s-4} (1 - s(u)) \cdot (1 - s(u + 3)) \cdot s_A(u + 4) \cdot s(u + 4) du, \quad (S25)$$

where

$$C_{S,3} = \int_{u=0}^{t_s-4} (1 - s(u)) \cdot (1 - s(u + 3)) du. \quad (S26)$$

For an asymptomatic individual, the probability that both BD Veritor and RT-PCR test produce a positive on day three of quarantine is

$$Q_{A,4} = \frac{1}{C_{A,3}} \int_{u=0}^{t_e} (1 - s(u)) \cdot (1 - s(u + 3)) \cdot s_A(u + 4) \cdot s(u + 4) du, \quad (\text{S27})$$

where

$$C_{A,3} = \int_{u=0}^{t_e} (1 - s(u)) \cdot (1 - s(u + 3)) du. \quad (\text{S28})$$

The probability that both BD Veritor and RT-PCR test produce a positive later in quarantine will be weighted more towards individuals that will never exhibit symptoms, as individuals entering quarantine who are soon-to-be symptomatic are more likely to have exhibited symptoms by the end of the quarantine. Specifying the proportion of asymptomatic infections to be p_A and the duration of the incubation period to be t_s , the proportion of asymptomatic infections that will be on day t of quarantine is

$$\tilde{p}_A(t) = p_A \max \left\{ \frac{t_s - t}{t_s}, 0 \right\} + \left(1 - \max \left\{ \frac{t_s - t}{t_s}, 0 \right\} \right). \quad (\text{S29})$$

Therefore, the probability that both BD Veritor and RT-PCR test produce a positive on day t of quarantine is

$$Q_t = \left(1 - \tilde{p}_A(t) \right) Q_{S,t} + \tilde{p}_A(t) Q_{A,t}. \quad (\text{S30})$$

Estimation of offshore prevalence of SARS CoV-2

We adjusted our framework used to estimate the effective reproductive number during serial testing (Supplementary Material: Frequency of testing to reduce the effective reproduction number), to estimate whether there were any additional cases in the offshore environment not detected by the serial testing conducted. Specifically, we use the number of RA positives, number of RA false positives, and the FDA empirical diagnostic sensitivity and specificity for the BD Veritor RA test in this inference. Prior to arriving offshore for serial testing, all individuals were tested with RT-PCR on arrival and with test results returned within approximately 24 h. Asserting no transmission offshore (as no observed outbreak occurred during the observation period), any infected individual must have obtained a false-negative RT-PCR test result. As none of the individuals that tested positive exhibited symptoms at the time of testing, the analysis focuses on asymptomatic cases. We specify that any infected individual needs to be infectious after their release from quarantine (i.e. the 24 h delay from obtaining test results). Thus, these individuals must enter quarantine at least a day prior to the end of their infectiousness.

In the absence of individualized data, and to simplify the calculations, the test result on the second day of the screening offshore is independent of the test outcome on the first day of screening. Similarly, the test result on the third day is independent of the test results on the first and second days of screening offshore.

The number of true-positives requires the individual to first test positive with the rapid antigen test and then positive with RT-PCR. Specifying an uninfected individual, the probability of a double false-positive on a given testing day is

$$P_{U,j} = \left(1 - \zeta_A \right) \left(1 - \zeta_R \right), \quad (\text{S31})$$

where ζ_A is the specificity of the rapid antigen test and ζ_R is the specificity of RT-PCR.

Specifying an infected individual, the probability of having an RT-PCR confirmed RA test on the first test day is

$$P_{I,1} = \frac{1}{C} \int_{u=0}^{t_e - q} h(u) s_A(u + q + t_1) s(u + q + t_1) du, \quad (\text{S32})$$

where

$$h(u) = (1 - s(u)) \text{ and} \quad (\text{S33})$$

$$C = \int_{u=0}^{t_e - q} h(u) du. \quad (\text{S34})$$

The probability of having an RT-PCR-confirmed RA test on the second test day is

$$P_{I,2} = \frac{1}{C} \int_{u=0}^{t_e-q} h(u) s_A(u+q+t_2) s(u+q+t_2) du. \quad (S35)$$

The probability of having an RT-PCR-confirmed RA test on the third and final test day is

$$P_{I,3} = \frac{1}{C} \int_{u=0}^{t_e-q} h(u) s_A(u+q+t_3) s(u+q+t_3) du. \quad (S36)$$

The number of false-positive rapid antigen tests requires the individual to first test positive with the rapid antigen test and then negative with RT-PCR. Specifying an uninfected individual, the probability of a false-positive rapid antigen test on any test day is

$$F_{U,j} = (1 - \zeta_A) \zeta_R. \quad (S37)$$

Specifying an infected individual, the probability of a false-positive rapid antigen test on the first test day is

$$F_{I,1} = \frac{1}{C} \int_{u=0}^{t_e-q} h(u) s_A(u+q+t_1) (1 - s(u+q+t_1)) du. \quad (S38)$$

The probability of a false-positive rapid antigen test on the second test day is

$$F_{I,2} = \frac{1}{C} \int_{u=0}^{t_e-q} h(u) s_A(u+q+t_2) (1 - s(u+q+t_2)) du. \quad (S39)$$

The probability of a false-positive rapid antigen test on the third and final test day is

$$F_{I,3} = \frac{1}{C} \int_{u=0}^{t_e-q} h(u) s_A(u+q+t_3) (1 - s(u+q+t_3)) du. \quad (S40)$$

As the number of infected individuals in the screening population increases, the probability of obtaining a false-negative from the rapid antigen test in the cohort will decrease. Specifying an infected individual, the probability of a false-negative from the rapid antigen test on the first test day is

$$F_{N,1} = \frac{1}{C} \int_{u=0}^{t_e-q} h(u) (1 - s_A(u+q+t_1)) du. \quad (S41)$$

The probability of a false-positive rapid antigen test on the second test day is

$$F_{N,2} = \frac{1}{C} \int_{u=0}^{t_e-q} h(u) (1 - s_A(u+q+t_2)) du. \quad (S42)$$

The probability of a false-positive rapid antigen test on the third and final test day is

$$F_{N,3} = \frac{1}{C} \int_{u=0}^{t_e-q} h(u) (1 - s_A(u+q+t_3)) du. \quad (S43)$$

We have assumed that there was no transmission offshore. Thus, we need to minimize the probability of post-quarantine transmission for all the infected individuals. Specifying isolation after a positive RT-PCR test, the expected transmission from an asymptomatic infected individual under the specified offshore screening protocol

$$\begin{aligned} R = & \frac{1}{C} \int_{u=0}^{t_e-q} \int_{t=q}^{q+t_1} h(u) r(t+u) dt du + \frac{1}{C} \int_{u=0}^{t_e-q} \int_{t=q+t_1}^{q+t_2} h(u) m_1(u) r(t+u) dt du \\ & + \frac{1}{C} \int_{u=0}^{t_e-q} \int_{t=q+t_2}^{q+t_3} h(u) m_1(u) m_2(u) r(t+u) dt du \end{aligned} \quad (S44)$$

$$+ \frac{1}{c} \int_{u=0}^{t_e-q} \int_{t=q+t_3}^{\infty} h(u) m_1(u) m_2(u) m_3(u) r(t+u) dt du,$$

where

$$m_1(u) = 1 - s_A(u+q+t_1)s(u+q+t_1), \quad (\text{S45})$$

$$m_2(u) = 1 - s_A(u+q+t_2)s(u+q+t_2), \quad (\text{S46})$$

and

$$m_3(u) = 1 - s_A(u+q+t_3)s(u+q+t_3). \quad (\text{S47})$$

Over the course of the data collection, the number of participants on each testing day was not constant (e.g. 124 tested on day two, and 96 tested on day eight). To account for changes in the cohort size, we specify the same reduction in the number of infected individuals. Denoting the number of people tested in the i th round of testing by N_i and specifying T_i infected individuals, the number of infected individuals that have remained offshore is $\frac{N_i}{N_1} T_i$.

Specifying an initial T_U uninfected individuals and T_I infected individuals within the cohort for testing strategy j , the expected number of false-positives on testing day i is

$$F_{ij} = \frac{N_i}{N_1} (T_U F_{U,1} + T_I F_{I,1}), \quad (\text{S48})$$

and the number of RT-PCR confirmed positives is

$$P_{ij} = \frac{N_i}{N_1} (T_U P_{U,1} + T_I P_{I,1}). \quad (\text{S49})$$

and the probability of false-negatives from a rapid antigen test for those not detected in the screening on the first test day is expressed as

$$F_{N,i}^{N_i T_i (1-P_{I,i}-F_{I,i})/N_1}. \quad (\text{S50})$$

The likelihood of the outcomes that occurred on the first testing day is

$$L_{ij} = f(\bar{F}_i | F_i) \cdot f(\bar{P}_i | P_i) \cdot F_{N,i}^{N_i T_i (1-P_{I,i}-F_{I,i})/N_1} \quad (\text{S51})$$

where \bar{F}_i is the number of observed false-positives on the testing day i , \bar{P}_i is the number of RT-PCR confirmed positives on testing day i , and $f(X|Y)$ is the Poisson distribution with mean Y .

To estimate that the number of infected individuals that were offshore with a probability of onward transmission p for the two testing strategies—computed from the expected transmission as in Eq. S44—we maximized

$$\prod_{j=1}^2 (1-p_j)^{T_{ij}} \prod_{i=1}^3 L_{ij}. \quad (\text{S52})$$

The effectiveness of serial testing conducted offshore in identifying asymptomatic cases is

$$\min \left\{ \frac{\sum_{j=1}^2 \bar{P}_{ij}}{2}, 1 \right\}. \quad (\text{S53})$$

Figure S36. Estimated infectious cases offshore. Specifying a basic reproduction number of 3.2 and 100% infections being asymptomatic, an incubation period of 5.72 days, the distribution of (A) the estimated number of infectious individuals in the offshore environment (black bars) relative to the number of RT-PCR confirmed cases observed (red dashed line) and (B) the estimated distribution of the effectiveness of the serial testing strategy conducted offshore based on the three cases detected in the offshore environment (black circles) and the approximated continuous distribution of the effectiveness of the serial testing (gray line). The continuous distribution was approximated using an interpolation function.

(3) The results are based on a logistic model for agreement between the RA and PCR tests over time. However, there are plenty of cases in the supplement where the logistic model does not appear to fit well, or where the model extrapolates very far past the point where any data is available at all. What is the motivation for specifying a logistic model? How can we be confident that it provides reliable extrapolations? One point of misspecification appears to be that RA tests have lower sensitivity in the very early stages of infection, which isn't captured by having a single (sign-constrained) coefficient. This might cause some issues with estimation, where the fit is a flat line (over time) because the model can't capture non-monotonic behavior.

Response: We agree with the reviewer that there are several cases where the linear logistic regression model does not fit the data well at all. Many of these time points for the percent positive agreement are based on six or fewer samples, which can create a large amount of uncertainty around that time point. As per the comment from Reviewer #2, we now integrate uncertainty around the percent positive agreement curves for each of the rapid antigen tests.

The motivation for specifying a logistic model for the percent positive agreement was to ensure that the percent positive agreement was bounded between zero and one in all parameter space. We restricted the logistic model to be linear to limit unexpected behavior in the percent positive agreement during extrapolation. We agree that the extrapolation is unreliable, and without additional data further into the disease time course these extrapolations cannot be validated. Thus, assuming the monotonic decrease after the viral peak was the most advantageous approach, as attempting to capture the non-monotonic behavior in some of the data could lead to over-parameterization and even more unreliable extrapolation. One could integrate a 'general' component for all RA tests into the fitting of the linear logistic model to inform

this decline in sensitivity/PPA, but this approach would be at the expense of losing/masking the individuality of each of the tests.

The misspecification of not having a lower sensitivity during the early stages of infection stems from the absence of percent positive agreement data during the incubation period. To overcome the absence of data, a mapping was constructed between the infectivity profile and the percent positive agreement to infer the percent positive agreement during the incubation period. Thus, not attaining the low sensitivity early on would be attributed to the non-rapid decline in the percent positive agreement after symptom onset.

I also have some more minor comments:

(1) The basic reproduction number assumed (2.5) seems a little out-of-date, given that delta is likely to be the predominant strain. The authors note that their qualitative conclusions will not be impacted by changes to R_0 , which is likely true, but the recommendations about quarantine lengths certainly will be. A sensitivity analysis of some of the key results to a higher value of R_0 would improve the paper.

Response: We have now integrated a sensitivity analysis surrounding the effects of R_0 on the expected post-quarantine transmission for the different tests and quarantine durations, as well as the impact it has on the frequency of serial testing in order to maintain $RE < 1$.

Added text to Methods:

We conducted scenario analyses to determine the impact of (i) the incubation period; (ii) reduced diagnostic sensitivity of RA tests for low-levels of infectivity; (iii) proportion of asymptomatic infections and the basic reproduction number; and (iv) the RT-PCR diagnostic sensitivity curve on the onward transmission after quarantine and during serial testing.

To evaluate the robustness of our results to a potentially longer duration of incubation than 4.4 days⁴¹, we evaluated the impact of an alternate incubation period of 5.72 days and the infectivity profile associated with that longer duration³⁷. A positive RA test indicates active infection, suggesting that there is a substantial concentration of virus within the sample site to return a positive. Studies have indicated that SARS-CoV-2 cannot be successfully cultured 10 days after symptom onset⁴⁴. Thus, it is possible at some stages of the disease that a RA test will only return a negative while RT-PCR still yields a positive. There is also uncertainty in the inferred diagnostic sensitivity in the later stages of disease, as the PPA was extrapolated past the last recorded data point for all RA tests. To address whether differences in the results of these tests among low levels of infectivity make a difference to our findings, we imposed a threshold on the level of infectivity such that the RA test will only return a negative. If the infectivity at the time of the RA tests is below the infectivity at 10 days post-symptom onset then the RA test returns a negative result (**Fig. S42**). We examined a stricter threshold based on the infectivity at 5.6 days post-symptom onset (where there is 1% infectivity remaining). To assess the sensitivity of our results to variation in the proportion of asymptomatic infection and the basic reproduction number, we conducted a grid analysis with values ranging from 0.1–0.95 and 1.05–5, respectively. Furthermore, we evaluated an alternative model for the temporal RT-PCR diagnostic sensitivity curve, substituting a log-Student's- t distribution for the log-Normal distribution used in the baseline analysis.

Added text to Results:

To examine the effect of the proportion of infections that are asymptomatic (p_A) and the basic reproduction number (R_0) on quarantine duration and the frequency of serial testing, we conducted a two-way sensitivity analysis for the five commonly used RA tests. For varying values of p_A and R_0 , we determined the minimum quarantine duration that results in equivalent or lower probability of PQT than

that computed for the RA test conducted on exit from a seven-day quarantine under the baseline parameterization ($p_A = 0.351$, $R_0 = 3.2$). The minimum quarantine required was positively associated with both the parameters (Fig. S37A–S41A). As R_0 decreases, the frequency of serial testing required to maintain $R_E < 1$ becomes increasingly sensitive to changes in p_A (Fig. S37B–S41B). As p_A increases for a specified R_0 , we found that more frequent serial testing should be conducted to maintain $R_E < 1$ (Fig. S37–S41).

Added text in Discussion:

The effectiveness of quarantine and serial testing strategies is also dependent on the effective reproduction number, which in turn is influenced by disease interventions. For example, a reduction in the effective reproduction number would occur during vaccine rollout when immunity increases in the population. We evaluated quarantine and serial testing strategies under the assumption that individuals self-isolate upon symptom onset, resulting in an R_E of 2.4 when a basic reproduction number of 3.2 was specified. Organizations, institutions, and localities will likely implement multiple disease-control measures, resulting in idiosyncratic R_E . Higher R_E or lower levels of self-isolation upon symptom onset will require more stringent testing strategies than indicated here.

Quantitatively, our results are dependent on epidemiological context. For example, breakthrough cases have been observed to less frequently be symptomatic and less frequently experience severe disease compared to unvaccinated cases^{69,70}. As illustrated in our scenario analyses, the equivalent quarantine durations to RT-PCR and effective frequency of serial testing could increase due to asymptomatic infections despite a reduction in the reproduction number. Considering community prevalence instead of transmission, investment in a testing strategy with greater specificity might be a more prudent approach in low-prevalence settings, whereas a testing strategy with high sensitivity becomes more important in communities with high prevalence. Nevertheless, the heterogeneity of diagnostic sensitivity of tests that we have demonstrated is independent of the context-specific basic reproduction number. We expect the relative performance of RA tests in comparison to RT-PCR to remain intact despite manifest differences in R_E .

Added Figures to Supplementary Material:

Figure S37. The impact of asymptomatic infection and basic reproduction number on quarantine duration and frequency of serial testing for BinaxNOW. Specifying an incubation period of 4.4 days and RT-PCR diagnostic sensitivity represented by a log-Normal distribution (A) the quarantine duration for a RA test on exit (blue and red gradient) that has an equivalent or lower probability of post-quarantine transmission than the baseline scenario (35.1% asymptomatic infection and a basic reproduction number of 3.2) for a 7-day quarantine (black dot) and (B) the frequency of serial testing required to maintain the effective reproduction number below one (black and white gradient; baseline indicated by yellow dot) for asymptomatic infections ranging from 10% to 95% and a

basic reproduction number ranging from 1.05 to 5.

Figure S38. The impact of asymptomatic infection and basic reproduction number on quarantine duration and frequency of serial testing for Sofia. Specifying an incubation period of 4.4 days and RT-PCR diagnostic sensitivity represented by a log-Normal distribution (A) the quarantine duration for a RA test on exit (blue and red gradient) that has an equivalent or lower probability of post-quarantine transmission than the baseline scenario (35.1% asymptomatic infection and a basic reproduction number of 3.2) for a 7-day quarantine (black dot) and (B) the frequency of serial testing required to maintain the effective reproduction number below one (black and white gradient; baseline indicated by yellow dot) for asymptomatic infections ranging from 10% to 95% and a basic reproduction number ranging from 1.05 to 5.

Figure S39. The impact of asymptomatic infection and basic reproduction number on quarantine duration and frequency of serial testing for BD Veritor. Specifying an incubation period of 4.4 days and RT-PCR diagnostic sensitivity represented by a log-Normal distribution (A) the quarantine duration for a RA test on exit (blue and red gradient) that has an equivalent or lower probability of post-quarantine transmission than the baseline scenario (35.1% asymptomatic infection and a basic reproduction number of 3.2) for a 7-day quarantine (black dot) and (B) the frequency of serial testing required to maintain the effective reproduction number below one (black and white gradient; baseline indicated by yellow dot) for asymptomatic infections ranging from 10% to 95% and a

basic reproduction number ranging from 1.05 to 5.

Figure S40. The impact of asymptomatic infection and basic reproduction number on quarantine duration and frequency of serial testing for LumiraDx (AN). Specifying an incubation period of 4.4 days and RT-PCR diagnostic sensitivity represented by a log-Normal distribution, **(A)** the quarantine duration for a RA test on exit (blue and red gradient) that has an equivalent or lower probability of post-quarantine transmission than the baseline scenario (35.1% asymptomatic infection and a basic reproduction number of 3.2) for a 7-day quarantine (black dot), and **(B)** the frequency of serial testing required to maintain the effective reproduction number below one (black and white gradient; baseline indicated by yellow dot) for asymptomatic infections ranging from 10% to 95% and a basic reproduction number ranging from 1.05 to 5.

Figure S41. The impact of asymptomatic infection and basic reproduction number on quarantine duration and frequency of serial testing for CareStart (AN). Specifying an incubation period of 4.4 days and RT-PCR diagnostic sensitivity represented by a log-Normal distribution, **(A)** the quarantine duration for a RA test on exit (blue and red gradient) that has an equivalent or lower probability of post-quarantine transmission than the baseline scenario (35.1% asymptomatic infection and a basic reproduction number of 3.2) for a 7-day quarantine (black dot), and **(B)** the frequency of serial testing required to maintain the effective reproduction number below one (black and white gradient; baseline indicated by yellow dot) for asymptomatic infections ranging from 10% to 95% and a

basic reproduction number ranging from 1.05 to 5.

(2) What role is being served by the RA test on entry to quarantine, as opposed to a test only at the end? This wasn't immediately clear.

Response: We have now clarified the role served by the RA test conducted on entry in both the Results and Discussion, connecting the modeling results and the empirical data

Added text to Methods:

For quarantine durations varying from one to 14 days, we compared the probability of PQT when performing a single RT-PCR test on exit to a RA test on exit or RA tests on both entry and exit. The objective of the additional RA test on entry to the one on exit is to compensate for the reduced diagnostic sensitivity of a single RA test compared to RT-PCR.

Added text to Discussion:

Supplementing an exit RA test with one upon entry to quarantine could allow RA tests to outperform a single RT-PCR test on exit for longer quarantine durations. Specifically, for more than 50% of RA tests considered, conducting a RA test on both entry and exit from quarantine produced a lower probability of post-quarantine transmission than an RT-PCR test on exit for quarantine durations up to 14 days. As prevalence in the community increases, having at least one false-negative in a cohort becomes more probable. Conducting a RA test on entry in addition to the one on exit mitigates the reduced sensitivity of a single test, thereby reducing the probability of releasing a still-infectious case from quarantine. Testing on alternative days to quarantine entry could increase the diagnostic sensitivity of the testing strategy even more than adding a test on entry. If logistically and financially feasible, diagnostic sensitivity can be increased further by testing on additional days. At a higher cost than a RA test, multiple RT-PCR tests could also be conducted in quarantine to increase the probability of identifying a case. Long quarantine durations have minimal practicality for entities trying to minimize interruptions to operations. By adding a less costly RA test on entry to the exit test, the quarantine duration could be reduced, substantially mitigating loss of productivity attributed to quarantine.

(3) What is the rationale for using the incubation period distribution provided by Qin et al instead of Lauer et al?

Response: Our initial rationale for using the incubation period estimated by Qin et al was to maintain consistency with our published analysis on optimizing quarantine durations. As per the comment from Reviewer #2 suggesting an adjustment towards analysis more reflective of the Delta variant, we now use an incubation period of 4.4 days in our primary analysis ⁵ and an incubation period of 5.72 days specified by Ashcroft et al ² as a scenario analysis in the Supplementary material.

Added text to Methods:

We conducted scenario analyses to determine the impact of (i) the incubation period; (ii) reduced diagnostic sensitivity of RA tests for low-levels of infectivity; (iii) proportion of asymptomatic infections and the basic reproduction number; and (iv) the RT-PCR diagnostic sensitivity curve on the onward transmission after quarantine and during serial testing.

⁵ Zhang et al: Transmission Dynamics of an Outbreak of the COVID-19 Delta Variant B.1.617.2 — Guangdong Province, China, May–June 2021, doi:10.46234/ccdcw2021.148

To evaluate the robustness of our results to a potentially longer duration of incubation than 4.4 days⁴¹, we evaluated the impact of an alternate incubation period of 5.72 days and the infectivity profile associated with that longer duration³⁷. A positive RA test indicates active infection, suggesting that there is a substantial concentration of virus within the sample site to return a positive. Studies have indicated that SARS-CoV-2 cannot be successfully cultured 10 days after symptom onset⁴⁴. Thus, it is possible at some stages of the disease that a RA test will only return a negative while RT-PCR still yields a positive. There is also uncertainty in the inferred diagnostic sensitivity in the later stages of disease, as the PPA was extrapolated past the last recorded data point for all RA tests. To address whether differences in the results of these tests among low levels of infectivity make a difference to our findings, we imposed a threshold on the level of infectivity such that the RA test will only return a negative. If the infectivity at the time of the RA tests is below the infectivity at 10 days post-symptom onset then the RA test returns a negative result (**Fig. S42**). We examined a stricter threshold based on the infectivity at 5.6 days post-symptom onset (where there is 1% infectivity remaining). To assess the sensitivity of our results to variation in the proportion of asymptomatic infection and the basic reproduction number, we conducted a grid analysis with values ranging from 0.1–0.95 and 1.05–5, respectively. Furthermore, we evaluated an alternative model for the temporal RT-PCR diagnostic sensitivity curve, substituting a log-Student's-*t* distribution for the log-Normal distribution used in the baseline analysis.

Added text to Results:

Compared to a base-case incubation period of 4.4 days, an alternative incubation period of 5.72 days with an infectivity profile that peaks 2.4 days later (**Fig. S35**) yielded somewhat contrasting results (**Table S5 vs Table S3**). In this alternative scenario, a sole RT-PCR test on exit from quarantines of at least six days outperformed all RA tests conducted on entry to and exit from quarantine (**Fig. S30**). This improved performance of RT-PCR for the alternative incubation period can be attributed to a lower average probability of identifying a case in the incubation period (0.487, 95% CrI: 0.421–0.521; vs 0.704, 95% CrI: 0.609–0.724) and to the greater proportion of transmission occurring after the incubation period (44.9% vs 37.1%). With the reduced ability in general to detect a case early in the disease time course, the addition of a RA test upon entry into quarantine provides only a marginal reduction to the PQT. For six of the 18 RA tests, the alternative incubation period necessitates an increase in frequency of testing (from every three days to every two days) to maintain $R_E < 1$ (**Table S5 vs Table S3**).

Reviewer #2 (Remarks to the Author):

This manuscript examines testing, isolation (of cases) and quarantine (of non-cases and non-symptomatic individuals) strategies in order to reduce the risk of SARS CoV-2 transmission, primarily within occupational settings. The authors use a model of infectivity averted, translated to an effective R measure, to assess PCR (assumed 1 day turnaround) and 18 Rapid Antigen tests (no turnaround time) in 1) quarantine and testing strategies, 2) repeat testing strategies, and 3) to examine expected numbers of false positive from different testing frequencies and technologies. The paper uses data on the percent positive agreement post symptom onset with PCR for a range of RA tests in order to obtain test sensitivity curves, using data from manufacturers but also presenting data from empirical screening studies (for the more limited number of tests for which this was available). PCR positivity is estimated with data from Hellewell et al, with some modifications to their method. A second study inferring PCR positivity is used as a sensitivity analysis. The study found that the transmission equivalent of a 7-day quarantine with PCR on exit policy could be broadly achieved using RA tests, with some variability among them (required quarantines of 6-11 days across the 18 tests). Assuming a 24 hr turnaround delay for diagnosing and isolating cases for PCR tests, the study found serial testing with tests taken every 6 days to be sufficient to keep $R < 1$ for PCR, and every 6-8 days for RA tests, with long PCR test return delays shifting favour towards RA tests. However, more intensive testing strategies using RAs also led to more false positives.

Overall, the approach of the work is sound and the qualitative conclusions, that lack of delay in obtaining RA test results as compared to PCR could at least or more than make up for lower overall sensitivity of the tests, are in line with the findings of other studies (eg Larremore et al <https://www.ncbi.nlm.nih.gov/pmc/articles/PMC7325181/>). The other paper I am aware of examining multiple RA tests (Pickering et al) does not go on to consider the implications for transmission within testing interventions. This paper could therefore constitute a helpful addition to the literature on choice of testing technology. It is useful that the paper also present empirical data on a test screening intervention for offshore workers, as well as modelling analyses. However, the use of a longer incubation period estimate than that seen in many other studies potentially is puzzling (mean 8.3 days compared to 5-6 days) and has implications for the findings because I think it affects the analysis of RA test sensitivity relative to PCR and the infectivity profile. I think it might lead to longer intervals between tests being found adequate to keep $R < 1$ (along with a low estimate of R_0). The estimate used is from a single study from very early in the epidemic and I cannot see a justification or rationale given for this choice compared to the lower estimates found in systematic review studies. I think it would therefore be helpful for the authors to give a strong justification for why this estimate is used and to conduct sensitivity analyses as to the effect on their findings. The authors should be commended for using a second estimate of the PCR test sensitivity curve, and this looks more like what I would expect for PCR relative to RA test sensitivity. My other concern is that, as I understand it, there is relatively little heterogeneity in infection progression incorporated across individuals (eg in the form of incubation periods or infectivity profiles resulting) and the findings are not expressed with any uncertainty or variability.

Major points

1. It was not clear to me why the authors preferred an incubation period of 8.29 days, from Qin et al. This estimate, from very early in the pandemic, is higher than that of many other studies, typically 5-6 days, and it is higher than the mean incubation period found in systematic reviews. I think if the authors choose to use this longer estimate it needs to be strongly justified given its importance to the analysis. Ideally they could conduct some sensitivity analyses to assess the extent to which findings might vary given assumptions about the distribution of the incubation period. I have not formally reviewed estimates of the incubation period, but other systematic reviews seem to put it significantly lower than what the authors use, eg: <https://www.acpjournals.org/doi/10.7326/m20-0504> 5.1 days (95% CI, 4.5 to 5.8 days) (also early 2020 data) <https://systematicreviewsjournal.biomedcentral.com/articles/10.1186/s13643-021-01648-y> The mean incubation period ranged from 5.2 (95% CI 4.4 to 5.9) to 6.65 days (95% CI 6.0 to 7.2) (up to dec 2020) <https://bmcinfectdis.biomedcentral.com/articles/10.1186/s12879-021-05950-x> the weighted pooled mean incubation period of COVID-19 was 6.5 (95%CI: 5.9–7.1) days (up to July 2020) <https://bmjopen.bmj.com/content/10/8/e039652> (5.8 (95% CI 5.0 to 6.7) days) (up to April 2020)

Response: We thank the reviewer for this important feedback on the duration of the incubation period. As per the comment from the reviewer suggesting an adjustment of R_0 towards analysis more reflective of the Delta variant, we now use an incubation period of 4.4 days⁶ in our primary analysis and an incubation period of 5.72 days specified by Ashcroft et al² as a scenario analysis.

These two scenarios have distinct infectivity profiles and RT-PCR curves which lead to qualitatively different performances of adding an entry RA test to an exit test. We specifically comment on the reason for this difference in the Results and the public health implications in the Discussion.

Added text to Methods:

We conducted scenario analyses to determine the impact of (i) the incubation period; (ii) reduced diagnostic sensitivity of RA tests for low-levels of infectivity; (iii) proportion of asymptomatic infections and the basic reproduction number; and (iv) the RT-PCR diagnostic sensitivity curve on the onward transmission after quarantine and during serial testing.

To evaluate the robustness of our results to a potentially longer duration of incubation than 4.4 days⁴¹, we evaluated the impact of an alternate incubation period of 5.72 days and the infectivity profile associated with that longer duration³⁷. A positive RA test indicates active infection, suggesting that there is a substantial concentration of virus within the sample site to return a positive. Studies have indicated that SARS-CoV-2 cannot be successfully cultured 10 days after symptom onset⁴⁴. Thus, it is possible at some stages of the disease that a RA test will only return a negative while RT-PCR still yields a positive. There is also uncertainty in the inferred diagnostic sensitivity in the later stages of disease, as the PPA was extrapolated past the last recorded data point for all RA tests. To address whether differences in the results of these tests among low levels of infectivity make a difference to our findings, we imposed a threshold on the level of infectivity such that the RA test will only return a negative. If the infectivity at the time of the RA tests is below the infectivity at 10 days post-symptom onset then the RA test returns a negative result (**Fig. S42**). We examined a stricter threshold based on the infectivity at 5.6 days post-symptom onset (where there is 1% infectivity remaining). To assess the sensitivity of our results to variation in the proportion of asymptomatic infection and the basic reproduction number, we conducted a grid analysis with values ranging from 0.1–0.95 and 1.05–5, respectively. Furthermore, we evaluated an alternative model for the temporal RT-PCR diagnostic sensitivity curve, substituting a log-Student's-*t* distribution for the log-Normal distribution used in the baseline analysis.

Added text to Results:

Compared to a base-case incubation period of 4.4 days, an alternative incubation period of 5.72 days with an infectivity profile that peaks 2.4 days later (**Fig. S35**) yielded somewhat contrasting results (**Table S5 vs Table S3**). In this alternative scenario, a sole RT-PCR test on exit from quarantines of at least six days outperformed all RA tests conducted on entry to and exit from quarantine (**Fig. S30**). This improved performance of RT-PCR for the alternative incubation period can be attributed to a lower average probability of identifying a case in the incubation period (0.487, 95% CrI: 0.421–0.521; vs 0.704, 95% CrI: 0.609–0.724) and to the greater proportion of transmission occurring after the incubation period (44.9% vs 37.1%). With the reduced ability in general to detect a case early in the disease time course, the addition of a RA test upon entry into quarantine provides only a marginal reduction to the PQT. For six of the 18 RA tests, the alternative incubation period necessitates an increase in frequency of testing (from every three days to every two days) to maintain $R_E < 1$ (**Table S5 vs Table S3**).

Added text to the Discussion:

The frequency of serial testing to maintain $R_E < 1$ was robust under alternative infectivity curves and incubation periods for the majority of the RA tests, with reductions in R_E being consistent with previous studies^{8,9,17}. However, the combined change in the infectivity profile and incubation period influenced the utility of RA tests in quarantine. When there is a low probability of identifying a case in the incubation period and more transmission after the incubation period, an RT-PCR on exit from quarantine reduces PQT more than a RA test on both entry and exit for quarantines of six days or longer duration. With the emergence of Omicron, the US CDC has shortened the recommended 7-day quarantine to a 5-day quarantine after exposure, followed by five days of strict mask use⁷⁵. Omicron is estimated to have an incubation period of about three days^{76,77}, roughly two and a half days shorter than the estimated

incubation period of the original pandemic virus (i.e., our alternative scenario). The infectivity profile of Omicron has not yet been determined. Preliminary evidence indicates that the viral load peaks between three to six days after diagnosis or symptom onset when infected with the Omicron variant.^{78,79} This later peak in viral load relative to prior variants could indicate a shift toward a greater proportion of transmission occurring after symptom onset. Under these conditions, we would hypothesize that RA tests will be less effective at identifying cases during a 5-day quarantine compared to RT-PCR. Diagnostic sensitivity of the RA tests has remained fairly stable as SARS-CoV-2 has evolved⁸⁰⁻⁸², but could, in principle, change with new emerging variants⁷². Our scenario analyses suggests that epidemiological characteristics of the disease (e.g. incubation period, infectivity profile, proportion of asymptomatic transmission) impact testing strategies more than moderate changes in diagnostic sensitivity.

As I understand it from the supplementary material, the longer assumed incubation time distribution (Qin et al) will also be reflected in the estimate of the infectivity profile because the authors state that they use the Qin et al incubation period distribution in their estimation of the diagnostic sensitivity curve of the RT-PCR test from the Hellewell et al data. They then state that this diagnostic sensitivity is constrained to peak at the same time as infectivity over the course of infectious age (which does make sense assuming both are linked to viral load in the naso-pharyngeal tract). Could the authors clarify whether this means that the infectivity profile that they then obtain tends to also peak relatively late, compared to other studies, and to give a generation interval that is relatively longer on average? This would presumably mean that longer intervals between tests in a serial testing strategy would be found to be more effective?. Could the authors comment on how their findings about the frequency of tests required to maintain $R < 1$ compare to other studies, eg Larremore et al 2021 (although I appreciate it is difficult to make different studies directly comparable with different assumptions)?

Response: The reviewer is correct that changes in the duration of the incubation period will be reflected in the infectivity profile, and influences the timing of peak infectiousness. The duration of the incubation period will impact the duration of quarantine and frequency of serial testing differently. The reviewer correctly pointed out that a shorter incubation period than 8.29 days will require more frequent serial testing, but will also lead to a shorter specified quarantine. Based on the comments from both reviewer, we now use a much shorter incubation period of 4.4 days for the base case (reflective of the Delta variant) and 5.72 days in a scenario analysis. As a result, we are no longer using the previous infectivity profile that was presented in the initial submission. However, we now illustrate the peak times of the infectivity profiles used in the analysis (Supplementary Figure 35) and the corresponding peak in RT-PCR diagnostic sensitivity (Supplementary Figure 24). As per the request from the reviewer, we also discuss our findings about the frequency of tests required to maintain $R < 1$ compared to other studies. Following text was added to the discussion section to address reviewers' comments:

Added text to the Discussion:

The frequency of serial testing to maintain $R_E < 1$ was robust under alternative infectivity curves and incubation periods for the majority of the RA tests, with reductions in R_E being consistent with previous studies^{8,9,17}.

I think that the long estimate for the incubation period also affects the extent to which RA tests are assumed to agree with PCR post-symptom since time since infection (Fig 1), because the

only measurements for agreement were relative to symptom onset, which in this study is placed relatively late in infection. I think that if the incubation period was assumed to have a mean of approximately 5 days, the decline of RA test sensitivity post symptom onset would have occurred earlier in the infectious age, and so is also responsible for the better looking RA sensitivity relative to PCR in the later stages of infection, than I would expect. My expectations are based on analyses which estimate RA test sensitivity stratified by CT value/viral load (eg as in https://www.liverpool.ac.uk/media/livacuk/research/Mass_testing_evaluation.pdf). Because PCR is more sensitive than RA tests at lower viral loads, as the authors themselves do mention, I would have expected a slightly later rise and faster decline in RA test sensitivity relative to PCR than shown in Figure 1.

Response: We thank the reviewer for raising this issue about the impact of the incubation period on the temporal diagnostic sensitivity of the RA tests. As the reviewer points out, the data used to inform the agreement between RA and RT-PCR tests is relative to the time symptom onset. With the absence of percent positive agreement data during the incubation period, we must infer the percent positive agreement. To make this inference, we construct a one-to-one mapping between the infectivity profile after symptom onset and the percent positive agreement to determine. This mapping and the percent positive agreement post-onset will dictate the rate of rise in the diagnostic sensitivity of this test. As both reviewers commented, a negative RA test is most apt in the late stages of disease. Therefore, based on reviewers feedback, we now assess this aspect by conducting a scenario analysis where a threshold is imposed as to when the RA test will only return a negative test result.

Added text to the Methods:

We conducted scenario analyses to determine the impact of (i) the incubation period; (ii) reduced diagnostic sensitivity of RA tests for low-levels of infectivity; (iii) proportion of asymptomatic infections and the basic reproduction number; and (iv) the RT-PCR diagnostic sensitivity curve on the onward transmission after quarantine and during serial testing.

To evaluate the robustness of our results to a potentially longer duration of incubation than 4.4 days⁴¹, we evaluated the impact of an alternate incubation period of 5.72 days and the infectivity profile associated with that longer duration³⁷. A positive RA test indicates active infection, suggesting that there is a substantial concentration of virus within the sample site to return a positive. Studies have indicated that SARS-CoV-2 cannot be successfully cultured 10 days after symptom onset⁴⁴. Thus, it is possible at some stages of the disease that a RA test will only return a negative while RT-PCR still yields a positive. There is also uncertainty in the inferred diagnostic sensitivity in the later stages of disease, as the PPA was extrapolated past the last recorded data point for all RA tests. To address whether differences in the results of these tests among low levels of infectivity make a difference to our findings, we imposed a threshold on the level of infectivity such that the RA test will only return a negative. If the infectivity at the time of the RA tests is below the infectivity at 10 days post-symptom onset then the RA test returns a negative result (**Fig. S42**). We examined a stricter threshold based on the infectivity at 5.6 days post-symptom onset (where there is 1% infectivity remaining). To assess the sensitivity of our results to variation in the proportion of asymptomatic infection and the basic reproduction number, we conducted a grid analysis with values ranging from 0.1–0.95 and 1.05–5, respectively. Furthermore, we evaluated an alternative model for the temporal RT-PCR diagnostic sensitivity curve, substituting a log-Student's-*t* distribution for the log-Normal distribution used in the baseline analysis.

Added text to the Results:

We examined a scenario in which if the level of infectivity at the time of testing was less than the infectivity 10 days after symptom onset then the RA test only returns a negative result. For this specified threshold in RA test diagnostic sensitivity, we observed no change in the results (Table S6). To explore

the impact of a narrower period in which a RA test can return a positive result, we imposed a higher threshold (i.e., the level of infectivity on 5.6 days after symptom onset). This higher threshold exhibited a moderate effect on the optimal testing strategies for some of the more effective RA tests. Specifically, the duration of quarantine needed to be extended by one day for four RA tests (CareStart (NS); Celltrion DiaTrust; LumiraDX (AS); Simoa) to achieve a probability of PQT equivalent or lower than a 7-day quarantine with an RT-PCR test on exit. In the context of serial testing, five RA tests (Clip COVID; Liaison (AS); Samoa; Sofia; Sofia 2 Flu + SARS) required a shorter time interval between tests by one day (i.e., more frequent testing; Table S6). Other than these observations, our results remained consistent with those obtained in our baseline analysis (Table S6).

Added text to the Discussion:

All of the PPA data for RA tests was gathered subsequent to symptom onset. This absence could be concerning because the early sensitivity of RA tests is a crucial aspect of their utility. Inaccurate characterization of early-phase diagnostic sensitivity would affect the results for entry testing, quarantines shorter than the incubation period, and the effectiveness of frequent serial testing. However, we were able to address the absence of PPA data in the incubation period by constructing a mapping between the infectivity and the PPA.

Most PPA data was gathered within the week following symptom onset, with some RA test datasets extending to later time points. This dearth of data later in the time course leads to higher uncertainty regarding late-disease diagnostic sensitivity. However, the relatively high uncertainty regarding the sensitivity of some RA tests in late disease only has a limited effect on our findings (affecting estimates of quarantine duration and frequency of testing by at most a day), due to the low infectiousness and consequently limited transmission that occurs late in disease. Because all results presented here relate to transmission, the higher probability that a RA test produces a false-negative result (compared with RT-PCR)^{29,71} in the late stages of disease—when the individual may no longer be infectious—matters far less than their higher probability of false-positive result during early stages of the disease. Even with the RA tests having distinct PPA trajectories from other RA tests—especially late in disease—the estimated quarantine durations and frequencies of serial testing were relatively robust among the 18 RA tests (differentiating at most a day). Therefore, we expect little to no change in these estimates as additional information about the diagnostic sensitivity over the disease course for each RA test emerges.

Added Table and Figure to the Supplementary material:

Table S6. The change in the required quarantine durations and serial testing frequencies when there is a specified threshold in which the rapid antigen test can return a positive test result relative to the results in Table S1 for the maximum likelihood estimates.

Rapid antigen test	Threshold based on infectivity at 5.6 days after symptom onset			Threshold based on infectivity at 10 days after symptom onset		
	Exit test ^a	Entry and exit test ^a	Frequency ^b	Exit test ^a	Entry and exit test ^a	Frequency ^b
BD Veritor ^{d,e}	0	0	0	0	0	0
BinaxNOW ^{e,f}	0	0	0	0	0	0
BinaxNOW ^{g,f}	0	0	0	0	0	0
BinaxNOW ^{h,f}	0	0	0	0	0	0

CareStart ^{e,f}	0	0	0	0	0	0
CareStart ^{e,i}	0	1	0	0	0	0
CareStart ^{f,g}	0	0	0	0	0	0
CareStart ^{f,h}	0	0	0	0	0	0
Celltrion DiaTrust ^{e,i}	0	1	0	0	0	0
Clip COVID ^{e,f}	0	0	- 1	0	0	0
Ellume ^{e,j}	0	0	0	0	0	0
Liaison ^{e,f}	0	0	- 1	0	0	0
Liaison ^{e,i}	0	0	0	0	0	0
LumiraDX ^{e,f}	0	1	0	0	0	0
LumiraDX ^{e,i}	0	0	0	0	0	0
Omnia ^{e,f}	0	0	0	0	0	0
SCoV-2 Ag Detect ^{e,f}	0	0	0	0	0	0
Simoa ^{e,i}	0	1	- 1	0	0	0
Sofia ^{e,f}	0	0	- 1	0	0	0
Sofia ^{h,f}	0	0	0	0	0	0
Sofia ^{g,f}	0	0	0	0	0	0
Sofia 2 Flu+SARS ^{e,f}	0	0	- 1	0	0	0
Status COVID-19/Flu ^{e,i}	0	0	0	0	0	0
Vitros ^{e,i}	0	0	0	0	0	0

^a Quarantine durations that are equivalent or better than a 7-day quarantine with an RT-PCR test conducted 24 h before exit.

^b The minimum required testing frequency for serial testing such that the effective reproductive number is less than one.

^c The probability of at least one false positive in a two-week period of serial testing under the minimum required testing frequency.

^d Peer-reviewed

^e Data from EUA submission

^f Anterior nasal swab

^g Data from community testing

^h Combined data from EUA submission and community testing

ⁱ Nasopharyngeal swab

^j Mid-turbinate swab

Figure S42. Temporal diagnostic sensitivity of a rapid antigen test when specifying a threshold day for returning a positive test. Specifying an incubation period of 4.4 days, the temporal diagnostic sensitivity of an RT-PCR test (black line), the BD Veritor rapid antigen test with no cut-off (red dashed line), the BD Veritor rapid antigen test with a threshold day specified by the level of infectivity 10 days after symptom onset (solid red line), and the BD Veritor rapid antigen test with a cut-off specified by the level of infectivity 5.6 days after symptom onset (solid yellow line).

The test sensitivity curve obtained in the sensitivity analysis using the Wells et al data, Figure S21, looks more like what I would expect for RA compared to RT-PCT sensitivity. The authors had modified the original Hellewell analysis and used the long Qin et al incubation period in the modified analysis - was this not also the case for the Wells curve, where the peak in PCR test sensitivity occurs earlier and is more consistent with a viral load, and therefore infectivity profile that peaks earlier than that assumed in the main paper analysis?

Response: The analysis using the two diagnostic sensitivity curves use the same infectivity profile. The difference between the two diagnostic sensitivity curves is the data used to inform them. From the Wells et al curve, the curve was constructed from diagnostic sensitivity data after symptom onset from Miller et al (patients presenting to the hospital) where the baseline curve was constructed from both pre-symptomatic and

post-symptomatic test results from screened healthcare workers (Hellewell et al). The qualitative difference is that the diagnostic sensitivity from Hellewell et al declines more rapidly after symptom onset compared to the curve obtained from Miller et al. This more rapid decline dampens the absolute difference between RA sensitivity and RT-PCR sensitivity shown for the Hellewell et al curve compared to that constructed from the Miller et al data. In both diagnostic sensitivity curves, the relative difference between RA and RT-PCR are the same. We note that after the revisions, the diagnostic sensitivity is only based on the Hellewell data. Three different diagnostic sensitivity curves were constructed by using a different incubation period and a different functional form of the diagnostic sensitivity curve. We have also added the infectivity profiles and RT-PCR curves for these three scenarios in the Supplementary material.

The average temporal infectivity curve. Specifying a basic reproduction number of 3.2 and 35.1% of infections being asymptomatic, the average infectivity curve for a known time of infection under no self-isolation upon symptom onset (black) and perfect isolation upon symptom onset (yellow line) for (A) an incubation period of 4.4 days (resulting in 2.4 secondary infections, yellow fill), and (B) an incubation period of 5.72 days (resulting in 2.3 secondary infections, yellow fill).

Figure S24: Diagnostic sensitivity of RT-PCR. The estimated diagnostic sensitivity of RT-PCR tests (black stars) informed by data from Hellewell et al⁹ for (A) an incubation period of 4.4 days (dashed vertical line) and the temporal RT-PCR diagnostic sensitivity represented by a log-Normal distribution, (B) an incubation period of 4.4 days (dashed vertical line) and the temporal RT-PCR diagnostic sensitivity represented by a log-Student's *t* distribution, (C) an incubation period of 5.72 days (dashed vertical line) and the temporal RT-PCR diagnostic sensitivity represented by a log-Normal distribution and (D) a comparison of the three temporal RT-PCR diagnostic sensitivity curves relative to the time of symptom onset (dashed vertical line)

2. There is no variation or uncertainty expressed for the estimated testing and quarantining strategies, whereas in practice we would expect heterogeneity in the progression of infections (incubation period, infectious periods, duration of infectiousness, and therefore test sensitivity over infection) for different individuals. Could the authors explain why no variability is shown?

Response: With the application of these control efforts being applied at the population level, the effectiveness of these interventions is most informative and is best reflected by the average characteristics of disease in the population. One could then examine the effects of uncertainty in the average on the duration of quarantine and frequency of testing. We now conduct a scenario analysis of the incubation period, considering an incubation period of 4.4 days and 5.72 days. We found that the results for surrounding serial testing are robust to this moderate change, while we obtained qualitatively different results for our comparison of RA test and RT-PCR tests in quarantine. We discuss these differences and implications of these results in detail. In addition, we now include uncertainty in our estimates.

There is limited data surrounding the percent positive agreement data for RA test to RT-PCR and based on the reviewers later comment about the use of different diagnostic sensitivity curves, we now construct uncertainty for the percent positive agreement of rapid antigen tests with RT-PCR, RT-PCR diagnostic sensitivity and test specificity. We have also conducted extensive scenario analysis surrounding the percent of

asymptomatic infections, the basic reproductive number, the duration of the incubation period (and corresponding infectivity curve), the functional form of the diagnostic sensitivity curve, and consider a window in which a RA test can return a positive result.

Added text to the Methods:

We conducted scenario analyses to determine the impact of (i) the incubation period; (ii) reduced diagnostic sensitivity of RA tests for low-levels of infectivity; (iii) proportion of asymptomatic infections and the basic reproduction number; and (iv) the RT-PCR diagnostic sensitivity curve on the onward transmission after quarantine and during serial testing.

To evaluate the robustness of our results to a potentially longer duration of incubation than 4.4 days⁴¹, we evaluated the impact of an alternate incubation period of 5.72 days and the infectivity profile associated with that longer duration³⁷. A positive RA test indicates active infection, suggesting that there is a substantial concentration of virus within the sample site to return a positive. Studies have indicated that SARS-CoV-2 cannot be successfully cultured 10 days after symptom onset⁴⁴. Thus, it is possible at some stages of the disease that a RA test will only return a negative while RT-PCR still yields a positive. There is also uncertainty in the inferred diagnostic sensitivity in the later stages of disease, as the PPA was extrapolated past the last recorded data point for all RA tests. To address whether differences in the results of these tests among low levels of infectivity make a difference to our findings, we imposed a threshold on the level of infectivity such that the RA test will only return a negative. If the infectivity at the time of the RA tests is below the infectivity at 10 days post-symptom onset then the RA test returns a negative result (**Fig. S42**). We examined a stricter threshold based on the infectivity at 5.6 days post-symptom onset (where there is 1% infectivity remaining). To assess the sensitivity of our results to variation in the proportion of asymptomatic infection and the basic reproduction number, we conducted a grid analysis with values ranging from 0.1–0.95 and 1.05–5, respectively. Furthermore, we evaluated an alternative model for the temporal RT-PCR diagnostic sensitivity curve, substituting a log-Student's-*t* distribution for the log-Normal distribution used in the baseline analysis.

Added text to the Results:

Compared to a base-case incubation period of 4.4 days, an alternative incubation period of 5.72 days with an infectivity profile that peaks 2.4 days later (**Fig. S35**) yielded somewhat contrasting results (**Table S5 vs Table S3**). In this alternative scenario, a sole RT-PCR test on exit from quarantines of at least six days outperformed all RA tests conducted on entry to and exit from quarantine (**Fig. S30**). This improved performance of RT-PCR for the alternative incubation period can be attributed to a lower average probability of identifying a case in the incubation period (0.487, 95% CrI: 0.421–0.521; vs 0.704, 95% CrI: 0.609–0.724) and to the greater proportion of transmission occurring after the incubation period (44.9% vs 37.1%). With the reduced ability in general to detect a case early in the disease time course, the addition of a RA test upon entry into quarantine provides only a marginal reduction to the PQT. For six of the 18 RA tests, the alternative incubation period necessitates an increase in frequency of testing (from every three days to every two days) to maintain $R_E < 1$ (**Table S5 vs Table S3**).

Added text to the Discussion:

The frequency of serial testing to maintain $R_E < 1$ was robust under alternative infectivity curves and incubation periods for the majority of the RA tests, with reductions in R_E being consistent with previous studies^{8,9,17}. However, the combined change in the infectivity profile and incubation period influenced the utility of RA tests in quarantine. When there is a low probability of identifying a case in the incubation period and more transmission after the incubation period, an RT-PCR on exit from quarantine reduces PQT more than a RA test on both entry and exit for quarantines of six days or longer duration. With the emergence of Omicron, the US CDC has shortened the recommended 7-day quarantine to a 5-day quarantine after exposure, followed by five days of strict mask use⁷⁵. Omicron is estimated to have an incubation period of about three days^{76,77}, roughly two and a half days shorter than the estimated

incubation period of the original pandemic virus (i.e., our alternative scenario). The infectivity profile of Omicron has not yet been determined. Preliminary evidence indicates that the viral load peaks between three to six days after diagnosis or symptom onset when infected with the Omicron variant.^{78,79} This later peak in viral load relative to prior variants could indicate a shift toward a greater proportion of transmission occurring after symptom onset. Under these conditions, we would hypothesize that RA tests will be less effective at identifying cases during a 5-day quarantine compared to RT-PCR. Diagnostic sensitivity of the RA tests has remained fairly stable as SARS-CoV-2 has evolved^{80–82}, but could, in principle, change with new emerging variants⁷². Our scenario analyses suggests that epidemiological characteristics of the disease (e.g. incubation period, infectivity profile, proportion of asymptomatic transmission) impact testing strategies more than moderate changes in diagnostic sensitivity.

Example Figure added to the Supplementary Material

Figure S3. The diagnostic sensitivity curve and probability of post-quarantine transmission for BD Veritor. Specifying a 4.4-day incubation period, 35.1% of infections being asymptomatic, self-isolation upon symptom onset, (A) the RT-PCR diagnostic sensitivity curve (black) informed by data from Hellewell et al⁹ using a log-Normal distribution and logistic regression model for the diagnostic sensitivity of the rapid antigen test (red, solid line) informed by percent positive agreement data (red triangles); (B) the probability of post-quarantine transmission for a RT-PCR test conducted 24 h before exit from quarantine (black stars), a rapid antigen test on exit (filled red triangles), as well as a rapid antigen test conducted on both entry to and exit from quarantine (open red triangles, dashed line); (C) the RT-PCR diagnostic sensitivity curve informed by data from Hellewell et al⁹ using a log-Student's *t* distribution and logistic regression model for the

diagnostic sensitivity of the rapid antigen test; (D) the probability of post-quarantine transmission for a RT-PCR test conducted 24 h before exit from quarantine, a rapid antigen test on exit, as well as a rapid antigen test conducted on both entry to and exit from quarantine. Specifying an 5.72-day incubation period, 35.1% of infections being asymptomatic, self-isolation upon symptom onset, (E) the RT-PCR diagnostic sensitivity curve informed by data from Hellewell et al ⁹ using a log-Normal distribution and logistic regression model for the diagnostic sensitivity of the rapid antigen test; and (F) the probability of post-quarantine transmission for a RT-PCR test conducted 24 h before exit from quarantine, a rapid antigen test on exit, as well as a rapid antigen test conducted on both entry to and exit from quarantine. The colored area denotes the 2.5 and 97.5 percentile from 1,000 samples of the RT-PCR diagnostic sensitivity curve and the rapid antigen tests percent positive agreement curve.

Example revised Table in the Supplementary Material

Table S3. Specifying a 4.4-day incubation period, a log-Normal distribution for the temporal RT-PCR diagnostic sensitivity, a basic reproductive number of 3.2, and 35.1% of infections being asymptomatic, the required quarantine durations, serial testing frequencies, and probabilities of false-positives associated with the serial testing frequency.

Rapid antigen test	Quarantine required		Serial testing required	
	Exit test ^a	Entry and exit test ^a	Frequency ^b	Prob. of a false positive ^c
BD Veritor ^{d,e}	8 (8–8)	7 (7–8)	2 (2–2)	0.0389 (0.00845–0.152)
BinaxNOW ^{e,f}	8 (8–8)	7 (7–7)	2 (2–2)	0.104 (0.0428–0.21)
BinaxNOW ^{g,f}	8 (8–8)	7 (7–7)	2 (2–3)	0.0474 (0.0264–0.0751)
BinaxNOW ^{h,f}	8 (8–8)	7 (7–7)	2 (2–2)	0.0558 (0.0364–0.0833)
CareStart ^{e,f}	8 (8–8)	7 (7–7)	2 (2–2)	0.00675 (0.00407–0.33)
CareStart ^{e,i}	8 (8–8)	6 (6–7)	3 (2–3)	0.0355 (0.00716–0.188)
CareStart ^{f,g}	8 (8–8)	7 (7–7)	2 (2–2)	0.117 (0.0771–0.17)
CareStart ^{f,h}	8 (8–8)	7 (7–7)	2 (2–2)	0.112 (0.0739–0.164)
Celltrion DiaTrust ^{e,i}	8 (8–8)	6 (6–7)	3 (2–3)	0.0488 (0.00846–0.26)
Clip COVID ^{e,f}	8 (8–8)	7 (6–7)	3 (2–3)	0.00451 (0.00315–0.144)
Ellume ^{e,j}	8	6	3	0.141

	(8–8)	(6–7)	(2–3)	(0.0628–0.357)
Liaison ^{e,f}	8 (8–8)	7 (6–7)	3 (2–3)	0.00451 (0.00325–0.175)
Liaison ^{e,i}	8 (8–8)	7 (6–7)	2 (2–3)	0.0575 (0.00863–0.221)
LumiraDX ^{e,f}	8 (8–8)	6 (6–7)	3 (2–3)	0.155 (0.0767–0.374)
LumiraDX ^{e,i}	8 (8–8)	6 (6–7)	3 (2–3)	0.108 (0.0473–0.28)
Omnia ^{e,f}	8 (8–8)	7 (7–7)	2 (2–2)	0.00675 (0.00419–0.478)
SCoV-2 Ag Detect ^{e,f}	8 (8–8)	7 (7–7)	2 (2–2)	0.00675 (0.00387–0.0857)
Simoa ^{e,i}	8 (8–8)	6 (6–7)	3 (2–3)	0.00451 (0.00323–0.393)
Sofia ^{e,f}	8 (8–8)	7 (6–7)	3 (2–3)	0.00451 (0.00329–0.113)
Sofia ^{g,f}	8 (8–8)	7 (7–8)	2 (1–2)	0.109 (0.0637–0.241)
Sofia ^{h,f}	8 (8–8)	7 (7–7)	2 (2–2)	0.0944 (0.0586–0.145)
Sofia 2 Flu+SARS ^{e,f}	8 (8–8)	7 (6–7)	3 (2–3)	0.00451 (0.00308–0.15)
Status COVID-19/Flu ^{e,i}	8 (8–8)	7 (6–7)	2 (2–3)	0.00675 (0.00363–0.243)
Vitros ^{e,i}	8 (8–8)	7 (7–8)	2 (1–2)	0.00675 (0.00402–0.284)

^a Quarantine durations that are equivalent or better than a 7-day quarantine with an RT-PCR test conducted 24 h before exit.

^b The minimum required testing frequency for serial testing such that the effective reproductive number is less than one.

^c The probability of at least one false positive in a two-week period of serial testing under the minimum required testing frequency.

^d Peer-reviewed

^e Data from EUA submission

^f Anterior nasal swab

^g Data from community testing

^h Combined data from EUA submission and community testing

ⁱ Nasopharyngeal swab

^j Mid-turbinate swab

3. From the description of the available data on the % positive agreement with PCR of each RA test, it sounds like there was no data to inform the relationship between PCR and RA test sensitivity pre symptom onset and that this was inferred based on a mapping between the relative infectivity function and the percent positive agreement post symptom onset. The uncertainty over this period is understandable given the lack of data available, but should be stated as a limitation in the Discussion (especially because, rightly, test sensitivity early in infection is highlighted to be important).

Response: We agree with the reviewer that this point is important to highlight in the main text. We have added the following text to both the Methods and Discussion to highlight this limitation.

Added text in the Discussion:

All of the PPA data for RA tests was gathered subsequent to symptom onset. This absence could be concerning because the early sensitivity of RA tests is a crucial aspect of their utility. Inaccurate characterization of early-phase diagnostic sensitivity would affect the results for entry testing, quarantines shorter than the incubation period, and the effectiveness of frequent serial testing. However, we were able to address the absence of PPA data in the incubation period by constructing a mapping between the infectivity and the PPA.

Most PPA data was gathered within the week following symptom onset, with some RA test datasets extending to later time points. This dearth of data later in the time course leads to higher uncertainty regarding late-disease diagnostic sensitivity. However, the relatively high uncertainty regarding the sensitivity of some RA tests in late disease only has a limited effect on our findings (affecting estimates of quarantine duration and frequency of testing by at most a day), due to the low infectiousness and consequently limited transmission that occurs late in disease. Because all results presented here relate to transmission, the higher probability that a RA test produces a false-negative result (compared with RT-PCR)^{29,71} in the late stages of disease—when the individual may no longer be infectious—matters far less than their higher probability of false-positive result during early stages of the disease. Even with the RA tests having distinct PPA trajectories from other RA tests—especially late in disease—the estimated quarantine durations and frequencies of serial testing were relatively robust among the 18 RA tests (differentiating at most a day). Therefore, we expect little to no change in these estimates as additional information about the diagnostic sensitivity over the disease course for each RA test emerges.

4. The method used to obtain measures of RA test specificity (shown in table S4) seems to mean that RA tests can only have a specificity as good as, or lower than that assumed for RT-PCR, 99.90%. What is the justification for this? Could the authors comment on how their findings regarding RA test specificity compare with others, eg Public Health England (now UK Health Security Agency) estimated the specificity of the Innova RA test to be 99.97%, which is higher than that assumed for other RA tests in this study (<https://www.gov.uk/government/publications/lateral-flow-device-specificity-in-phase-4-post-marketing-surveillance/lateral-flow-device-specificity-in-phase-4-post-marketing-surveillance>).

Response: We thank the reviewer for raising this point for us to clarify our approach. The estimated specificity of the RA test in our analysis is based on the percent negative agreement with RT-PCR, assuming that the RT-PCR is the gold-standard in terms of specificity and sensitivity. This assumption is consistent with that of the study mentioned by the reviewer, where Innova RA test has a specificity of 99.97% and RT-PCR has an assumed specificity of 100%. We now explicitly state this assumption and its influence on the specificity of RA tests in the Methods.

We now integrate uncertainty into the specificity of these RA tests as well and present this uncertainty in the Supplementary Tables

Added text to Methods:

Defining RT-PCR to be the gold-standard for testing accuracy, the specificity of a RA test is estimated by the product of the percent negative agreement of the RA test with RT-PCR and the specificity of the RT-PCR test. This assumption provides a lower bound for the RA test specificity, as the RT-PCR result could have been a false-negative.

Revised table in Supplementary Material:

Table S7. The specificity of RT-PCR and the EUA rapid antigen tests and 95% credible interval constructed from 1000 samples.

Test	Fraction of RA tests in agreement with negative RT-PCR	Specificity (95% CrI)	Data Source	Reference
RT-PCR	N/A	99.90% (99.84%–99.95%) (12392/12404)	Community testing	8
BD Veritor ^a	212/213	99.43% (97.58%–99.91%)	EUA submission ^b	13
BinaxNOW ^a	338/343	98.45% (96.60%–99.43%)	EUA submission	14
BinaxNOW ^a	2004/2016	99.31% (98.91%–99.59%)	Community testing	15
BinaxNOW ^a	2342/2359	99.18% (98.75%–99.49%)	EUA submission and community testing	14,15
CareStart ^a	53/53	99.90% (94.40%–99.92%)	EUA submission	16
CareStart ^d	147/148	99.23% (96.50%–99.90%)	EUA submission	16
CareStart ^a	1243/1264	98.24% (97.38%–98.90%)	Community testing	17
CareStart ^a	1296/1317	98.31% (97.52%–98.89%)	EUA submission and community testing	16,17
Celltrion ^d	102/103	98.93% (95.06%–99.86%)	EUA submission	18
Clip COVID ^a	134/134	99.90% (97.52%–99.96%)	EUA submission	19
Ellume ^e	156/161	96.80% (93.11%–98.74%)	EUA submission	20

Liaison ^a	108/108	99.90% (96.84%–99.96%)	EUA submission	21
Liaison ^d	133/134	99.16% (96.16%–99.87%)	EUA submission	21
LumiraDx ^a	168/174	96.46% (92.83%–98.57%)	EUA submission	22
LumiraDx ^d	210/215	97.58% (94.73%–99.08%)	EUA submission	23
Omnia ^a	32/32	99.90% (90.58%–99.92%)	EUA submission	24
SCoV-2 Ag Detect ^a	257/257	99.90% (98.72%–99.95%)	EUA submission	25
Simoa ^d	38/38	99.90% (92.14%–99.92%)	EUA submission	26
Sofia ^a	179/179	99.90% (98.09%–99.96%)	EUA submission	27
Sofia ^a	1025/1041	98.37% (97.46%–99.03%)	Community testing	28
Sofia ^a	1204/1220	98.59% (97.77%–99.16%)	EUA submission and community testing	27,28
Sofia 2 Flu + SARS ^a	122/122	99.90% (97.55%–99.96%)	EUA submission	29
Status COVID-19/Flu ^d	76/76	99.90% (95.70%–99.96%)	EUA submission	30
VITROS ^d	75/75	99.90% (95.45%–99.97%)	EUA submission	31

^a Anterior nasal swab

^b Peer-reviewed EUA data

^c Calculated based on both symptomatic and asymptomatic individuals

^d Nasopharyngeal swab

^e Mid-turbinate swab

4. It was a strength of the paper that the authors used a second study to determine the PCR sensitivity curve (well et al) over time since infection and compared their findings.

Response: We thank the reviewer for this comment. With the changes made based on reviewer comments, we had to remove the specific sensitivity curve mentioned by the

reviewer, as the diagnostic sensitivity during the incubation period no longer reflected reality (as diagnostic sensitivity increased rapidly after infection). To maintain this strength of an alternative RT-PCR curve, we used a log-Student's t -distribution in a scenario analysis. This new alternative curve has a lower peak diagnostic sensitivity and a slower decline in diagnostic sensitivity compared to our baseline RT-PCR curve (similar properties to that from the Miller et al data). Also, the scenario analysis of an alternative incubation period (and corresponding distribution of the incubation period from a separate study) provides another alternative RT-PCR curve in which we evaluate the utility of RA tests in quarantine and serial testing. Finally, uncertainty analysis was conducted for all three scenarios examined in the manuscript to illustrate the impact of the RT-PCR diagnostic sensitivity curve on the extent of post-quarantine transmission and the minimal frequency of serial testing needed to maintain $R_E < 1$.

Added text to the Results:

Compared to the RT-PCR diagnostic sensitivity curve (log Normal) applied in the baseline analysis, an alternative functional form (log Student's t) for the temporal sensitivity yields a higher probability of detecting infection over a longer duration, but a lower probability of detecting a case at the peak of infection (**Fig. S24**). Under this alternative RT-PCR sensitivity curve, the results for the probability of PQT were largely similar to those in the baseline analysis (**Fig. 2 vs Fig. S25; Table S3 vs Table S4**). The primary difference for the alternative RT-PCR curve was that CareStart (NS), Celltrion DiaTrust, Ellume, LumiraDx (AS), and Simoa required a one-day longer quarantine period for testing on entry and exit from quarantine than their corresponding quarantine duration in the baseline (**Table S3 vs Table S4**). For serial testing, 10 of the 18 RA tests required testing every two days under the alternative RT-PCR curve instead of every three days to maintain R_E below one (**Table S3 vs Table S4**).

Added Figure to Supplementary Material

Figure S24: Diagnostic sensitivity of RT-PCR. The estimated diagnostic sensitivity of RT-PCR tests (black stars) informed by data from Hellewell et al⁹ for (A) an incubation period of 4.4 days (dashed vertical line) and the temporal RT-PCR diagnostic sensitivity represented by a log-Normal distribution, (B) an incubation period of 4.4 days (dashed vertical line) and the temporal RT-PCR diagnostic sensitivity represented by a log-Student's t distribution, (C) an incubation period of 5.72 days (dashed

vertical line) and the temporal RT-PCR diagnostic sensitivity represented by a log-Normal distribution and (D) a comparison of the three temporal RT-PCR diagnostic sensitivity curves relative to the time of symptom onset (dashed vertical line)

Example of added Figure to Supplementary Material

Figure S3. The diagnostic sensitivity curve and probability of post-quarantine transmission for BD Veritor. Specifying a 4.4-day incubation period, 35.1% of infections being asymptomatic, self-isolation upon symptom onset, (A) the RT-PCR diagnostic sensitivity curve (black) informed by data from Hellewell et al⁹ using a log-Normal distribution and logistic regression model for the diagnostic sensitivity of the rapid antigen test (red, solid line) informed by percent positive agreement data (red triangles); (B) the probability of post-quarantine transmission for a RT-PCR test conducted 24 h before exit from quarantine (black stars), a rapid antigen test on exit (filled red triangles), as well as a rapid antigen test conducted on both entry to and exit from quarantine (open red triangles, dashed line); (C) the RT-PCR diagnostic sensitivity curve informed by data from Hellewell et al⁹ using a log-Student's *t* distribution and logistic regression model for the diagnostic sensitivity of the rapid antigen test; (D) the probability of post-quarantine transmission for a RT-PCR test conducted 24 h before exit from quarantine, a rapid antigen test on exit, as well as a rapid antigen test conducted on both entry to and exit from quarantine. Specifying an 5.72-day incubation period, 35.1% of infections being asymptomatic, self-isolation upon symptom onset, (E) the RT-PCR diagnostic sensitivity curve informed by data from Hellewell et al⁹ using a

log-Normal distribution and logistic regression model for the diagnostic sensitivity of the rapid antigen test; and (F) the probability of post-quarantine transmission for a RT-PCR test conducted 24 h before exit from quarantine, a rapid antigen test on exit, as well as a rapid antigen test conducted on both entry to and exit from quarantine. The colored area denotes the 2.5 and 97.5 percentile from 1,000 samples of the RT-PCR diagnostic sensitivity curve and the rapid antigen tests percent positive agreement curve.

5. It is also a great strength of this study that empirical analyses of relative test performance, and of serial testing, are incorporated.

Response: We thank the reviewer for this compliment.

6. The value for R_0 of 2.5 is quite out of date for the Delta variant (or the previously dominant Alpha variant). I am very sympathetic to the fast-evolving epidemiological situation and what this means for keeping an analysis up-to-date, but I think the authors should at least discuss the implications of an R_0 higher than 2.5 on their findings.

Response: We have added the quantitative impact of R_0 on our results to the discussion point about the qualitative impact of R_0 . We also now integrated a sensitivity analysis surrounding the effects of R_0 on the expected post-quarantine transmission for the different tests and quarantine durations. Furthermore, we have updated all of our baseline parameterization to reflect the characteristics of the Delta variant.

Added text to Results:

To examine the effect of the proportion of infections that are asymptomatic (p_A) and the basic reproduction number (R_0) on quarantine duration and the frequency of serial testing, we conducted a two-way sensitivity analysis for the five commonly used RA tests. For varying values of p_A and R_0 , we determined the minimum quarantine duration that results in equivalent or lower probability of PQT than that computed for the RA test conducted on exit from a seven-day quarantine under the baseline parameterization ($p_A = 0.351$, $R_0 = 3.2$). The minimum quarantine required was positively associated with both the parameters (**Fig. S37A–S41A**). As R_0 decreases, the frequency of serial testing required to maintain $R_E < 1$ becomes increasingly sensitive to changes in p_A (**Fig. S37B–S41B**). As p_A increases for a specified R_0 , we found that more frequent serial testing should be conducted to maintain $R_E < 1$ (**Fig. S37–S41**).

Added text in Discussion:

The effectiveness of quarantine and serial testing strategies is also dependent on the effective reproduction number, which in turn is influenced by disease interventions. For example, a reduction in the effective reproduction number would occur during vaccine rollout when immunity increases in the population. We evaluated quarantine and serial testing strategies under the assumption that individuals self-isolate upon symptom onset, resulting in an R_E of 2.4 when a basic reproduction number of 3.2 was specified. Organizations, institutions, and localities will likely implement multiple disease-control measures, resulting in idiosyncratic R_E . Higher R_E or lower levels of self-isolation upon symptom onset will require more stringent testing strategies than indicated here.

Quantitatively, our results are dependent on epidemiological context. For example, breakthrough cases have been observed to less frequently be symptomatic and less frequently experience severe disease compared to unvaccinated cases^{69,70}. As illustrated in our scenario analyses, the equivalent quarantine durations to RT-PCR and effective frequency of serial testing could increase due to asymptomatic infections despite a reduction in the reproduction number. Considering community prevalence instead of transmission, investment in a testing strategy with greater specificity might be a more prudent approach in low-prevalence settings, whereas a testing strategy with high sensitivity becomes more important in communities with high prevalence. Nevertheless, the heterogeneity of diagnostic sensitivity of tests that

we have demonstrated is independent of the context-specific basic reproduction number. We expect the relative performance of RA tests in comparison to RT-PCR to remain intact despite manifest differences in R_E .

Added Figures to Supplementary Material:

Figure S37. The impact of asymptomatic infection and basic reproduction number on quarantine duration and frequency of serial testing for BinaxNOW. Specifying an incubation period of 4.4 days and RT-PCR diagnostic sensitivity represented by a log-Normal distribution (A) the quarantine duration for a RA test on exit (blue and red gradient) that has an equivalent or lower probability of post-quarantine transmission than the baseline scenario (35.1% asymptomatic infection and a basic reproduction number of 3.2) for a 7-day quarantine (black dot) and (B) the frequency of serial testing required to maintain the effective reproduction number below one (black and white gradient; baseline indicated by yellow dot) for asymptomatic infections ranging from 10% to 95% and a basic reproduction number ranging from 1.05 to 5.

Figure S38. The impact of asymptomatic infection and basic reproduction number on quarantine duration and frequency of serial testing for Sofia. Specifying an incubation period of 4.4 days and RT-PCR diagnostic sensitivity represented by a log-Normal distribution (A) the quarantine duration for a RA test on exit (blue and red gradient) that has an equivalent or lower probability of post-quarantine transmission than the baseline scenario (35.1% asymptomatic infection and a basic reproduction number of 3.2) for a 7-day quarantine (black dot) and (B) the frequency of serial testing

required to maintain the effective reproduction number below one (black and white gradient; baseline indicated by yellow dot) for asymptomatic infections ranging from 10% to 95% and a basic reproduction number ranging from 1.05 to 5.

Figure S39. The impact of asymptomatic infection and basic reproduction number on quarantine duration and frequency of serial testing for BD Veritor. Specifying an incubation period of 4.4 days and RT-PCR diagnostic sensitivity represented by a log-Normal distribution (A) the quarantine duration for a RA test on exit (blue and red gradient) that has an equivalent or lower probability of post-quarantine transmission than the baseline scenario (35.1% asymptomatic infection and a basic reproduction number of 3.2) for a 7-day quarantine (black dot) and (B) the frequency of serial testing required to maintain the effective reproduction number below one (black and white gradient; baseline indicated by yellow dot) for asymptomatic infections ranging from 10% to 95% and a basic reproduction number ranging from 1.05 to 5.

Figure S40. The impact of asymptomatic infection and basic reproduction number on quarantine duration and frequency of serial testing for LumiraDx (AN). Specifying an incubation period of 4.4 days and RT-PCR diagnostic sensitivity represented by a log-Normal distribution, (A) the quarantine duration for a RA test on exit (blue and red gradient) that has an equivalent or lower probability of post-quarantine transmission than the baseline scenario (35.1% asymptomatic infection and a basic reproduction number of 3.2) for a 7-day quarantine (black dot), and (B) the frequency of

serial testing required to maintain the effective reproduction number below one (black and white gradient; baseline indicated by yellow dot) for asymptomatic infections ranging from 10% to 95% and a basic reproduction number ranging from 1.05 to 5.

Figure S41. The impact of asymptomatic infection and basic reproduction number on quarantine duration and frequency of serial testing for CareStart (AN). Specifying an incubation period of 4.4 days and RT-PCR diagnostic sensitivity represented by a log-Normal distribution, (A) the quarantine duration for a RA test on exit (blue and red gradient) that has an equivalent or lower probability of post-quarantine transmission than the baseline scenario (35.1% asymptomatic infection and a basic reproduction number of 3.2) for a 7-day quarantine (black dot), and (B) the frequency of serial testing required to maintain the effective reproduction number below one (black and white gradient; baseline indicated by yellow dot) for asymptomatic infections ranging from 10% to 95% and a basic reproduction number ranging from 1.05 to 5.

7. Similarly, it would be helpful for the authors to explicitly reflect in the Discussion on how their findings translate to a setting with vaccination and past infection.

Response: We now expand on the discussion of the effects of R_0 on our results, to translate these results in a setting with vaccination and past infection.

Added text to the Discussion:

The effectiveness of quarantine and serial testing strategies is also dependent on the effective reproduction number, which in turn is influenced by disease interventions. For example, a reduction in the effective reproduction number would occur during vaccine rollout when immunity increases in the population. We evaluated quarantine and serial testing strategies under the assumption that individuals self-isolate upon symptom onset, resulting in an R_E of 2.4 when a basic reproduction number of 3.2 was specified. Organizations, institutions, and localities will likely implement multiple disease-control measures, resulting in idiosyncratic R_E . Higher R_E or lower levels of self-isolation upon symptom onset will require more stringent testing strategies than indicated here.

Quantitatively, our results are dependent on epidemiological context. For example, breakthrough cases have been observed to less frequently be symptomatic and less frequently experience severe disease compared to unvaccinated cases^{69,70}. As illustrated in our scenario analyses, the equivalent quarantine durations to RT-PCR and effective frequency of serial testing could increase due to asymptomatic infections despite a reduction in the reproduction number. Considering community prevalence instead of

transmission, investment in a testing strategy with greater specificity might be a more prudent approach in low-prevalence settings, whereas a testing strategy with high sensitivity becomes more important in communities with high prevalence. Nevertheless, the heterogeneity of diagnostic sensitivity of tests that we have demonstrated is independent of the context-specific basic reproduction number. We expect the relative performance of RA tests in comparison to RT-PCR to remain intact despite manifest differences in R_E .

8. For the analysis of how testing might shorten the required quarantine period, the authors are not clear about what the quarantine period they discuss is for, ie whom they expect to be quarantining, for what reason. This is important because it has implications for the likely infectious age of infected individuals as they enter quarantine, and this affects both test sensitivity and transmissibility. This needs to be clarified and differentiated between the different scenarios throughout the paper. For instance for a close contact of a case entering quarantine, their likely infectious age will be a function of the time between exposure to the case and the time that they are notified/traced. Whereas for offshore workers, infected individuals will presumably be random across their pre-symptomatic infectious age, (depending on symptom status and whether or not symptomatic individuals would be expected to already be quarantining). If the modelling is primarily to assist with the example of sending staff offshore, and when the randomly sampling of infectious age is more reasonable, this should be explicitly stated, rather than only mentioned as an example. (Though I would argue that because symptomatic individuals should be opting out anyway, you will tend to have more people earlier in their infections at the start of quarantine, than at the end).

Response: We now explicitly state that entry into quarantine is random and is not reflective of a close contact of a case identified through contact tracing entering quarantine.

Revised text in Introduction:

To determine when these RA tests can serve as a suitable alternative to the more costly and laborious RT-PCR tests, we calculated (i) their associated probabilities of post-quarantine transmission (PQT) for quarantine durations from one to 14 days with testing on exit or both entry and exit upon random entry into quarantine; (ii) their extents of onward transmission for serial testing conducted every day to every 14 days; and (iii) their associated probabilities of false-positives during serial testing.

Revised text in Methods:

Specifying 35.1% of infections are asymptomatic⁵¹ and isolation upon symptom onset, we quantified the effectiveness of quarantine and testing strategies in reducing PQT by calculating the probability of PQT for individuals entering quarantine randomly—and not identified through contact tracing—in the absence of symptoms⁴. Accounting for substantial variance in transmission among COVID-19 cases^{4,52-56}, we specified that secondary cases were negative-binomially distributed:

$$f(x|k, p) = \frac{\Gamma(k+x)}{\Gamma(k)\Gamma(x+1)} p^k (1 - p)^x,$$

with dispersion parameter $k = 0.25$ ^{4,56} and $p = k / (k + R)$ —such that the average number of secondary cases is equal to the expected PQT, denoted R . This value for the dispersion parameter is consistent with estimates from other studies.⁵²⁻⁵⁵ Accordingly, the probability of PQT was calculated as $1 - f(0|k, p)$.

9. Figure 1 shows the sensitivity of RT-PCR (and some of the RA tests, appearing to be 1 or very close to 1 at its peak. Is this correct? This is different to the original Hellewell et al paper in which PCR sensitivity peaks at approx. 80%. Or is the y axis scale intended to be relative to

PCR at its peak? If the latter, please indicate this more clearly in the figure labelling and/or notes.

Response: The peak is very close to 1, which is different from the original Hellewell et al paper in which PCR sensitivity peaks at 80%. Although the fitting of the RT-PCR curve is similar, the construction of the diagnostic sensitivity curve is entirely different. Hellewell et al show the mean diagnostic sensitivity at each time point calculated from the posterior samples of the times of infection from a logistic piecewise regression. Here, we illustrate a single maximum likelihood estimate of the diagnostic sensitivity curve for the entire course of infection. Our curve still has a larger peak than the maximum posterior estimated from Hellewell et al. but is compensated by a narrower peak when compared to Hellewell et al. We also feel that this larger peak is more reflective of the estimated sensitivity of RT-PCR test. Many studies (one of such mentioned by the reviewer ⁶) indicate that RT-PCR sensitivity is greater than 90%. This sensitivity would be reflective of an average sensitivity over a specified period of infection. Under the Hellewell et al curve, this average sensitivity for a single RT-PCR test will never exceed 80%. For the day centered at the time of symptom onset, the average diagnostic sensitivity is 91%.

We now also provide uncertainty intervals for the RT-PCR curve, integrating this into our analysis as well. From this uncertainty analysis, the peak diagnostic sensitivity is 100% (95% credible interval: 86.5%–100%). We have expanded on these differences in the supplement.

Added text in Supplementary Material:

Our inference of the temporal diagnostic sensitivity of RT-PCR over the course of disease differs from that of Hellewell et al ⁹. In our three scenarios in which the temporal diagnostic sensitivity of RT-PCR is determined, we estimate a greater peak in diagnostic sensitivity (100%; 95% Credible Interval [CrI]: 86.5%–100%) ; 87.2% (95% CrI: 69.2%–99.1%); and 96.4%; 95% CrI: 88.0%–99.7%) ;) than Hellewell et al ⁹ (77%; 95% CrI: 54–88%). The greater diagnostic sensitivity in the peak is compensated by a narrower peak when compared to the breadth of the peak from Hellewell et al ⁹.

⁶ Lateral flow device specificity in phase 4 (post-marketing) surveillance. (March 2021). UK Government. <https://www.gov.uk/government/publications/lateral-flow-device-specificity-in-phase-4-post-marketing-surveillance/lateral-flow-device-specificity-in-phase-4-post-marketing-surveillance>

Figure S24: Diagnostic sensitivity of RT-PCR. The estimated diagnostic sensitivity of RT-PCR tests (black stars) informed by data from Hellewell et al⁹ for (A) an incubation period of 4.4 days (dashed vertical line) and the temporal RT-PCR diagnostic sensitivity represented by a log-Normal distribution, (B) an incubation period of 4.4 days (dashed vertical line) and the temporal RT-PCR diagnostic sensitivity represented by a log-Student's *t* distribution, (C) an incubation period of 5.72 days (dashed vertical line) and the temporal RT-PCR diagnostic sensitivity represented by a log-Normal distribution and (D) a comparison of the three temporal RT-PCR diagnostic sensitivity curves relative to the time of symptom onset (dashed vertical line)

Table S10. The estimated and 95% high posterior credible interval for the temporal RT-PCR diagnostic sensitivity curve

Incubation period (days)	Functional form	Parameter determining time of peak sensitivity (K)	Parameter determining the shape of the curve (z)	Scaling parameter (C)	Peak diagnostic sensitivity
4.4 ¹⁰	log-Normal	1.81 (1.69–2)	0.822 (0.749–0.932)	8.95 (7.2–10.5)	1 (0.865–1)
4.4 ¹⁰	log-Student's t	0.322 (0.322–0.322)	9.99×10^7 ($52.2-10^8$)	2.18 (1.81–2.46)	0.872 (0.692–0.991)
5.72 ¹²	log-Normal	2.04 (1.97–2.13)	0.581 (0.52–0.658)	9.1 (7.52–10.4)	0.964 (0.88–0.997)

10. It is very good that the authors compared test sensitivity and specificity for both the company datasets and real-world data because one might expect these to differ. It would be useful when describing this on p10 to briefly describe who the external populations were and how they might vary or not compared to the target population (eg were they lay people self-swabbing, or swabbing conducted by a health professional?).

Response: Thanks for the feedback. As suggested we have added a brief description to highlight the characteristics of the real-world data used for comparison. This description of the external populations is provided in the Methods section where the data is first introduced.

Added text to the Methods:

With informed consent from the offshore oil workers, onshore paired testing was conducted by laboratory-based RT-PCR and RA testing on entry into quarantine (day 0), day three, and day four. For both RA and RT-PCR testing, swabbing within quarantine was conducted by medical personnel.

Added text to the Methods:

We compared the PPA datasets submitted to the U.S.A FDA with those obtained from independent studies that were conducted in a real-world setting. Specifically, we considered the independent studies for BinaxNOW⁴⁹, Carestart⁵⁰, and Sofia³². Both the BinaxNOW and Carestart studies were conducted at a community testing site, where the trained site collector obtained the samples^{49,50}. The study for Sofia was conducted in a university setting, where the samples informing the PPA of the RA test with RT-PCR were from the university in which medical professionals conducted the swabbing³².

Added text to the Results:

We conducted a comparative analysis for these available internal and external datasets. Internal datasets for both CareStart and Sofia were not significantly different from the external datasets, while they were significantly different for BinaxNow (likelihood ratio tests; **Table S9**).

Added text to Supplementary Information

Comparison of the internal and external percent positive agreement data sets

To determine if the internal (i.e., USA FDA) dataset differs significantly from the external (i.e. independently conducted outside a controlled setting), we used the likelihood ratio test. We calculated the log-likelihood for the linear logistic regression curve for the i) the internal dataset (L_I), ii) the external dataset (L_E), and iii) from combining the two datasets (L_{IE}). The internal and external datasets are significantly different from each other if

$$L_I + L_E - L_{IE} \geq \chi^2_{0.95, 2}/2,$$

where $\chi^2_{0.95, 2}$ is the 95th percentile of the Chi-squared distribution with two degrees of freedom.

Otherwise, if

$$L_I + L_E - L_{IE} < \chi^2_{0.95, 2}/2,$$

then the two datasets are not significantly different from each other.

Added Table to Supplementary Information

Table S9. Comparison of the internal and external percent positive agreement datasets for three rapid antigen tests

Rapid Antigen Test	Log-likelihood			Difference ^a	Are the datasets significantly different
	Internal	External	Combined		
BinaxNOW	- 83.64	- 27.53	- 124.62	13.45	Yes
CareStart (AS)	- 14.88	- 26.55	- 41.49	0.05	No
Sofia	- 4.38	- 19.79	- 26.72	2.54	No

^a The difference may not reflect the difference based on values presented in the table due to rounding in the presentation of the log-likelihood values

Minor points:

11. Abstract:

*Suggest changing COVID-19 infections in first sentence to SARS-CoV-2 infections.

*Suggest second sentence saying what is being compared (potential transmissibility?)

Response: We have incorporated the changes suggested for the abstract.

Revised text in Abstract:

Rapid antigen (RA) tests are being increasingly employed to detect SARS-CoV-2 infections in quarantine and surveillance.

Revised text in Abstract:

For 18 RA tests with emergency use authorization from the United States of America FDA and an RT-PCR test, we conducted a comparative analysis of the post-quarantine transmission, the effective reproduction number during serial testing, and the false-positive rates.

12. Introduction first paragraph : “For instance, previous studies have shown that a 14-day quarantine with no testing can safely be shortened to a seven-day quarantine if a nasopharyngeal RT-PCR test is conducted on exit from the quarantine 4–6, a reduction notified for widespread implementation by the Centers for Disease Control and Prevention (CDC)7.” : suggesting specifying that this is quarantine of a close contact, rather than isolation of a case (this distinction might not be clear to all readers).

Response: We have now revised the sentence for clarity as follows:

For instance, previous studies have shown that a 14-day quarantine with no testing for a close contact of a case can safely be shortened to seven days if an RT-PCR test is conducted on exit from the quarantine ⁴⁻⁶.

13. p2, final paragraph - it would be worth mentioning too that the proportion of false positives will vary with prevalence so the extent to which we are worried about false positives versus false negative will change over the course the epidemic. Also of concern could the behavioural effects of receiving a negative test (in case these might lead to a higher level of risk-taking than would have occurred with no test at al).

Response: Thank you for your suggestion. We have now added a paragraph in the discussion to highlight these points surrounding the impact of prevalence on false positives and false negatives

Added text to the Discussion:

As prevalence in the community increases, having at least one false-negative in a cohort becomes more probable

Added text to the Discussion:

Using RA tests can mitigate some of the challenges and costs of serial testing with RT-PCR. However, RA tests can increase the chance of producing a false positive. From an operational standpoint, sending numerous or essential employees home could be critically problematic due to false-positive results⁶⁷. Several factors drive the consequences of surveillance frequency in terms of the number of false positives and extent of transmission, including (i) the sensitivity and specificity of the test, (ii) the number of people tested, (iii) the frequency of testing, and (iv) the background prevalence of the disease. For example, in a low-prevalence setting the probability of obtaining at least one false-positive in a cohort increases geometrically with the number of people tested and similarly with the frequency of testing per individual. Less frequent RA tests will reduce the chances of a false-positive during serial testing in a low-prevalence community, but at the cost of an outbreak occurring upon missing an infectious individual ($R_E > 1$). Assiduous follow-up testing of identified potential cases that are isolated should be incorporated—especially remote offshore settings where false-positive cases may require extensive and costly measures such as evacuation. Given the higher specificity of RT-PCR using an initial RA test with an RT-PCR follow-up for any positive can substantially reduce the number of RT-PCR tests, especially in a low-prevalence setting⁶⁸. A challenge to follow-up with RT-PCR testing is not just the increased logistical and economic costs: false-negatives can result in reintroduction into the surveilled population and onward transmission. Multiple follow-up tests are likely necessary to investigate conflicting test results. Accordingly, there are tradeoffs between the risks of transmission, number of tests conducted, processing times, and underlying costs from false positives regardless of the test utilized in surveillance. Therefore, policy decision-makers can adapt the testing schemes based on the current status of the epidemic to a level of risk that they find acceptable.

Added text to the Discussion:

Considering community prevalence instead of transmission, investment in a testing strategy with greater specificity might be a more prudent approach in low-prevalence settings, whereas a testing strategy with high sensitivity becomes more important in communities with high prevalence.

14. p3 :”One of the critical control strategies identified early on was systematic quarantine and testing” -> could the authors clarify the circumstances of the quarantine they are investigating? Is this a quarantine prior to joining the workforce offshore? Or is this a quarantine if a close contact of someone who has tested positive in an offshore environment (or any environment)? As mentioned above, there are implications for where in their infectious age we might expect any quarantined cases’ infectious ages to be.

Response: We thank the reviewer for raising commenting on this point. After revisions, this text is no longer included in the manuscript.

15. P4: could the final paragraph of the introduction clarify that the analysis was specifically designed to inform quarantine and serial testing strategies for offshore workers in the extractive industry - this is implied but not so clearly stated.

Response: We have now clarified that the analysis specifically was designed to analyze quarantine durations for individuals randomly entering quarantine and serial testing strategies for various populations. We clarify further the relevance of the analysis to testing strategies for offshore workers and the importance of the empirical data presented.

Revised text in Introduction:

Here we construct the temporal diagnostic sensitivity curves for 18 RA tests that have received EUA using data on percent positive agreement (PPA) with an RT-PCR test and temporal diagnostic sensitivity of an RT-PCR test. To determine when these RA tests can serve as a suitable alternative to the more costly and laborious RT-PCR tests, we calculated (i) their associated probabilities of post-quarantine transmission (PQT) for quarantine durations from one to 14 days with testing on exit or both entry and exit upon random entry into quarantine; (ii) their extents of onward transmission for serial testing conducted every day to every 14 days; and (iii) their associated probabilities of false-positives during serial testing. We further evaluated the utility of RA tests using data collected from two offshore oil companies in the context of quarantine and serial testing within an industrial environment.

16. "With a negligible delay in turnaround time, RT-PCR tests can be performed just once every eight days." : please could a more specific figure be given instead of 'negligible'. Does this mean <24 hrs, 12 hrs, no delay?

Response: After revisions, this text has been removed.

17. Figure 2 caption: I think the RT-PCR test on exit line line was a solid black line with stars, not circles.

Response: We have now corrected the caption for Figure 2.

Figure 2. Probability of post-quarantine transmission. Specifying a negative-binomial distribution for expected post-quarantine transmission, 35.1% of infections being asymptomatic, a 24-h delay in obtaining RT-PCR test results, no delay in receiving rapid antigen test results, an incubation period of 4.4 days, self-isolation upon symptom onset, and the diagnostic sensitivity curve for RT-PCR based on data from Hellewell et al ¹¹, the probability of post-quarantine transmission when conducting an RT-PCR test only on exit (solid line; black stars) and the rapid antigen tests (dashed lines) LumiraDx (blue squares); Sofia (green diamonds); BinaxNOW (yellow triangles); and BD Veritor (red circles); and CareStart (purple hexagram) performed (A) on exit and (B) on both entry and exit; and the fraction of the 18 rapid antigen tests whose use conferred a lower probability of post-quarantine transmission than did an RT-PCR test conducted 24 h before exit from quarantine, when the rapid antigen test was conducted (C) on exit and (D) on both entry and exit. The error bars denote the 95% credible interval based on 1,000 samples of an RT-PCR diagnostic sensitivity curve and rapid antigen percent positive agreement curve conducted through importance sampling.

18. It makes sense to me that for short quarantines, RT-PCR test on exit strategies would be less effective than the RA tests as modelled because they are taken a day earlier than the RA tests, and the probability of testing positive at this point in time is lower than RA tests taken a day later? This then switches around 6 days of quarantine, after which the effect of the higher sensitivity of PCR later on in infection dominates - is that the intuition for why this finding is

observed? (Fig 2A). And then in Fig 2B, the relative lack of sensitivity of a single RA test result later in infection is made up for by the improvement in sensitivity via having taken a RA test on entry as well? It might be worth highlighting this intuition if so.

Response: We thank the reviewer for bringing this point to our attention. As per both reviewer comments, we now expand this point of the importance of the entry test for RA.

Added text to Methods

For quarantine durations varying from one to 14 days, we compared the probability of PQT when performing a single RT-PCR test on exit to a RA test on exit or RA tests on both entry and exit. The objective of the additional RA test on entry to the one on exit is to compensate for the reduced diagnostic sensitivity of a single RA test compared to RT-PCR.

Added text to Discussion:

For quarantine policies with an exit test, RT-PCR tests typically must be conducted at least a day prior to the end of quarantine. In contrast, RA tests can be used closer to the end of quarantine due to rapid turnaround times. As a result of increasing test sensitivities during short quarantines⁴, most RA tests conducted on exit outperformed RT-PCR tests for quarantine durations less than three days. Supplementing an exit RA test with one upon entry to quarantine could allow RA tests to outperform a single RT-PCR test on exit for longer quarantine durations. Specifically, for more than 50% of RA tests considered, conducting a RA test on both entry and exit from quarantine produced a lower probability of post-quarantine transmission than an RT-PCR test on exit for quarantine durations up to 14 days. As prevalence in the community increases, having at least one false-negative in a cohort becomes more probable. Conducting a RA test on entry in addition to the one on exit mitigates the reduced sensitivity of a single test, thereby reducing the probability of releasing a still-infectious case from quarantine. Testing on alternative days to quarantine entry could increase the diagnostic sensitivity of the testing strategy even more than adding a test on entry. If logistically and financially feasible, diagnostic sensitivity can be increased further by testing on additional days. At a higher cost than a RA test, multiple RT-PCR tests could also be conducted in quarantine to increase the probability of identifying a case. Long quarantine durations have minimal practicality for entities trying to minimize interruptions to operations. By adding a less costly RA test on entry to the exit test, the quarantine duration could be reduced, substantially mitigating loss of productivity attributed to quarantine. Therefore, RA tests can be suitable alternatives to a single RT-PCR, especially for short quarantines of one or two days.

19. Discussion : p14, about false-positives “and consideration should be devoted to appropriate follow-up testing of potential cases that are identified and isolated.” -> I think this should be modified to account for community prevalence at the time of testing - our relative concern about false positives might vary accordingly (although it might be greater in the example of offshore testing na potential need for evacuation, for instance, where the cost is higher). It is unclear to me how and whether the risk of a false positive from PCR was assessed? CHECK this for previous analyses.

Response: We have integrated the importance of community prevalence at the time of testing in the paragraph of false-positives and contrast it to the environment of the offshore environment (where prevalence is expected to be low due to prior control measures that employees underwent to arrive). We also clarify that the risk of false-positives from RT-PCR was assessed (Table S6 and Figure 4).

Text in Methodology:

Probability of a false-positive

With a specificity ζ_i for test i (**Table S7**) and testing every f days, we computed the average

probability that at least one false-positive occurred over a two-week period $P_f = \frac{1}{f} \sum_{j=1}^f 1 - \zeta_i^{\tau_j}$,

where τ_j is the number of tests to occur in the j^{th} two-week period since the start of serial testing. For each testing frequency f (i.e., the time between two consecutive tests), we investigated the sequence of test times $\{1, 1 + f, 1 + 2f, \dots, 1 + 13f\}$ that comprises all the unique testing patterns possible over a two-week period to calculate the average probability. Defining RT-PCR to be the gold standard for testing accuracy, the specificity of a RA test was estimated as the specificity of the RT-PCR test multiplied by the percent negative agreement of the RA test with RT-PCR. This calculation provides a lower bound for the RA test specificity, given the possibility of a false-negative RT-PCR test.

Revised text in Results:

As the frequency of serial testing increases, an additional consideration to the reduction of transmission is the concomitant probability of obtaining false positives. For the purposes of our evaluation, we define the serial-testing false-positive rate as the probability that one or more tests yield false positives over a two-week period at the minimum frequency of testing required to maintain $R_E < 1$. Our calculations revealed an inverse relationship between R_E and the probability of obtaining at least one false-positive result (**Fig. 4** and **Fig. S23**; $r = -0.475$ and $P < 0.001$). For a specified R_E , RT-PCR tests (with at most a 48-h delay) yielded a lower probability of false-positive results than eight of the 18 RA tests (**Fig. S23**). These eight higher false-positive RA tests include BD Veritor, BinaxNOW, and LumiraDx, which are among the five that are most frequently used (**Fig. 4**). The serial testing false-positive rate ranged from 0.0045–0.155 among the 18 RA tests (**Table S3**), whereas it was 0.013 (95% CrI: 0.004–0.022) for RT-PCR testing with a 24-h delay. Among the 18 RA tests, false-positive rates clustered into two groups: a group of eight whose serial testing false-positive rates (ranging from 0.0355–0.155) markedly exceeded RT-PCR (**Fig. S23, Table S3**), and 10 that exhibited markedly lower serial testing false-positive rates (but for which the upper bound of the 95% credible interval still exceeded the serial testing false-positive rate for RT-PCR with a 24-h delay; **Table S3**).

Added text to Discussion:

Using RA tests can mitigate some of the challenges and costs of serial testing with RT-PCR. However, RA tests can increase the chance of producing a false positive. From an operational standpoint, sending numerous or essential employees home could be critically problematic due to false-positive results⁶⁷. Several factors drive the consequences of surveillance frequency in terms of the number of false positives and extent of transmission, including (i) the sensitivity and specificity of the test, (ii) the number of people tested, (iii) the frequency of testing, and (iv) the background prevalence of the disease. For example, in a low-prevalence setting the probability of obtaining at least one false-positive in a cohort increases geometrically with the number of people tested and similarly with the frequency of testing per individual. Less frequent RA tests will reduce the chances of a false-positive during serial testing in a low-prevalence community, but at the cost of an outbreak occurring upon missing an infectious individual ($R_E > 1$). Assiduous follow-up testing of identified potential cases that are isolated should be incorporated—especially remote offshore settings where false-positive cases may require extensive and costly measures such as evacuation. Given the higher specificity of RT-PCR using an initial RA test with an RT-PCR follow-up for any positive can substantially reduce the number of RT-PCR tests, especially in a low-prevalence setting⁶⁸. A challenge to follow-up with RT-PCR testing is not just the increased logistical and economic costs: false-negatives can result in reintroduction into the surveilled population and onward transmission. Multiple follow-up tests are likely necessary to investigate conflicting test results. Accordingly, there are tradeoffs between the risks of transmission, number of tests conducted,

processing times, and underlying costs from false positives regardless of the test utilized in surveillance. Therefore, policy decision-makers can adapt the testing schemes based on the current status of the epidemic to a level of risk that they find acceptable.

20. Abstract: please specify the FDA, 'in the United States', to make the setting clear.

Response: We now specify the FDA in the United States.

Revised text in the Abstract:

For 18 RA tests with emergency use authorization from the United States of America FDA and an RT-PCR test, we conducted a comparative analysis of the post-quarantine transmission, the effective reproduction number during serial testing, and the false-positive rates.

REVIEWERS' COMMENTS:

Reviewer #1 (Remarks to the Author):

Thank you to the authors for the detailed response. All of my major comments have been addressed. I still think that the logistic form/extrapolation for the sensitivity of the RA test is concerning , but the new sensitivity analysis to a hard cutoff for 0% effectiveness of RA tests effectively addresses this issue because the cutoff occurs around the point where extrapolation becomes dubious, testing a pessimistic version of what the curve might be at that point. I'd encourage the authors to mention a connection along these lines, that the new sensitivity analysis also effectively tests the impact of the extrapolated sensitivity of RA tests being inaccurate.

Reviewer #1 (Remarks to the Author):

Thank you to the authors for the detailed response. All of my major comments have been addressed. I still think that the logistic form/extrapolation for the sensitivity of the RA test is concerning, but the new sensitivity analysis to a hard cutoff for 0% effectiveness of RA tests effectively addresses this issue because the cutoff occurs around the point where extrapolation becomes dubious, testing a pessimistic version of what the curve might be at that point. I'd encourage the authors to mention a connection along these lines, that the new sensitivity analysis also effectively tests the impact of the extrapolated sensitivity of RA tests being inaccurate.

Response: We have now included comments in our Discussion explaining that the precision of our profile of the sensitivity of the RA tests drops off significantly late in disease time course, before noting that this issue does not affect our findings:

Most PPA data was gathered within the week following symptom onset, with some RA test datasets extending to later time points. This dearth of data later in the time course leads to **low precision in the extrapolation of the PPA beyond the extant data points, and** high uncertainty regarding late-disease diagnostic sensitivity. **The late-disease diagnostic sensitivity of the RA tests extrapolated beyond the data points should be applied with caution in other contexts where it might have consequences.** However, the **extrapolations and** high uncertainty **in RA-test late-disease diagnostic sensitivity** has a limited effect on our findings (affecting estimates of quarantine duration and frequency of testing by at most a day), due to the low infectiousness and consequently limited transmission that occurs late in disease.

Because all results presented here relate to transmission, a higher probability that a RA test produces a false-negative result (compared with RT-PCR)^{29,72} in the late stages of disease—when the individual may no longer be infectious—matters far less than their higher probability of producing a false-positive result during early stages of the disease. RA tests exhibit distinct PPA trajectories, especially late in disease; however, the estimated quarantine durations and frequencies of serial testing were relatively robust among the 18 RA tests (differentiating at most a day). Therefore, we expect little to no change in these estimates as additional information about the diagnostic sensitivity over the disease course for each RA test emerges.